# Interaction between transcribing RNA polymerase and topoisomerase I prevents R-loop formation in *E. coli*

Dmitry Sutormin [1,2] ✉, Alina Galivondzhyan [1,3], Olga Musharova [1,4],
Dmitrii Travin[1,2], Anastasiia Rusanova[2], Kseniya Obraztsova[5,8],
Sergei Borukhov [5] & Konstantin Severinov [1,6,7] ✉

Bacterial topoisomerase I (TopoI) removes excessive negative supercoiling and is thought to relax DNA molecules during transcription, replication and other processes. Using ChIP-Seq, we show that TopoI of *Escherichia coli* (EcTopoI) is colocalized, genome-wide, with transcribing RNA polymerase (RNAP). Treatment with transcription elongation inhibitor rifampicin leads to EcTopoI relocation to promoter regions, where RNAP also accumulates. When a 14 kDa RNAP-binding EcTopoI C-terminal domain (CTD) is overexpressed, colocalization of EcTopoI and RNAP along the transcription units is reduced. Pull-down experiments directly show that the two enzymes interact in vivo. Using ChIP-Seq and Topo-Seq, we demonstrate that EcTopoI is enriched upstream (within up to 12-15 kb) of highly-active transcription units, indicating that EcTopoI relaxes negative supercoiling generated by transcription. Uncoupling of the RNAP:EcTopoI interaction by either overexpression of EcTopoI competitor (CTD or inactive EcTopoI Y319F mutant) or deletion of EcTopoI domains involved in the interaction is toxic for cells and leads to excessive negative plasmid supercoiling. Moreover, uncoupling of the RNAP:EcTopoI interaction leads to R-loops accumulation genome-wide, indicating that this interaction is required for prevention of R-loops formation.

An optimal level of DNA supercoiling is required for DNA replication and transcription[1,2], DNA compaction, efficient bulk segregation of chromosomes, and site-specific DNA recombination and repair[3,4]. Topoisomerases, a conserved and ubiquitous group of enzymes, contribute to and regulate the extent of DNA supercoiling and its other topological properties[5]. Topoisomerases are divided into two types: type I enzymes introduce a transient single-strand break into DNA, while type II enzymes introduce a transient double-strand break[5].

Topoisomerase I of *Escherichia coli* (EcTopoI, encoded by the *topA* gene) belongs to the A class of type I topoisomerases[5]. EcTopoI relaxes only negatively supercoiled DNA and is thought to maintain the steady-state level of supercoiling by compensating the activity of another topoisomerase—the DNA gyrase, a type IIA enzyme, which introduces negative supercoiling utilizing the energy of ATP hydrolysis[6–8]. Deletion of *topA* leads to rapid accumulation of suppressor mutations, mostly in genes encoding the DNA gyrase subunits. By reducing gyrase

[1]Skolkovo Institute of Science and Technology, Moscow 121205, Russia. [2]Institute of Gene Biology RAS, Moscow 119334, Russia. [3]Lomonosov Moscow States University, Moscow 119991, Russia. [4]Institute of Molecular Genetics, National Research Centre "Kurchatov Institute", Moscow 123182, Russia. [5]Department of Cell Biology and Neuroscience, Rowan University School of Osteopathic Medicine, Stratford, NJ 08084-1489, USA. [6]Center for Precision Genome Editing and Genetic Technologies for Biomedicine, Institute of Gene Biology, RAS, Moscow 119334, Russia. [7]Waksman University for Microbiology, Rutgers, NJ 08854, USA. [8]Present address: University of Pennsylvania, Perelman School of Medicine, Department of Medicine, Philadelphia, PA 19104, USA. ✉e-mail: D.A.Sutormin@gmail.com; severik@waksman.rutgers.edu

activity, these mutations balance the level of DNA supercoiling inside the cell[9,10]. Amplification of a chromosomal region containing the *parC* and *parE* genes encoding topoisomerase IV (TopoIV) is also frequently reported in *topA* null mutants[11–13]. Conversely, *topA* deletions complement growth and replication defects of temperature-sensitive (Ts) *gyrB* mutants at non-permissive temperatures[14].

Hypernegative supercoiling is a hallmark of *topA* mutants resulting from uncompensated gyrase activity[15]. It was proposed that hypernegative supercoiling leads to the stabilization of R-loops containing RNA-DNA heteroduplexes formed when nascent transcripts anneal to the template DNA strand upstream of the transcribing RNA polymerase (RNAP)[15–17]. Indeed, R-loops have been recently detected in *topA* mutants by dot-blots assays with an RNA:DNA hybrid-specific antibody[18]. Since one DNA strand is unpaired in the R-loop, a hub accumulating excessive negative supercoiling is created, leaving the nearby DNA less negatively supercoiled. Because such DNA is a substrate for DNA gyrase, which introduces more negative supercoiling, a positive feedback loop is created, leading to further accumulation of R-loops and additional negative supercoiling (negative supercoiling → R-loops → masking of negative supercoiling → increased gyrase activity → excessive negative supercoiling) (Supplementary Fig. 1)[19]. As a result, nascent transcripts in R-loops are degraded, which leads to rapid growth arrest[15].

Overexpression of RNase HI, an enzyme which degrades RNA in the R-loops[20,21] and should thus break the feedback loop, was reported to partially suppress the negative effects of a *topA* deletion[22,23], although this finding was disputed[24], while deletion of the RNase HI *rnhA* gene exacerbates the *topA* null phenotype[24,25]. Stabilized R-loops can also prime *oriC*-independent replication—a phenomenon called "constitutive stable DNA replication" (cSDR) initially observed in cells lacking RNase HI. It was demonstrated that cells lacking type I topoisomerases also exhibit cSDR, which is suppressed by overexpression of RNase HI[13,26]. Together, these data indicate that hypernegative supercoiling is the likely cause of severe growth defects of non-suppressed *topA* mutants[27].

The EcTopoI was shown to bind RNAP, and the interaction was mapped to the C-terminal portion of EcTopoI and the β′ subunit of RNAP[28]. The RNAP:TopoI interaction was also reported for mycobacteria (Banda, Cao, and Tse-Dinh, 2017) and *Streptococcus pneumoniae*[29]. It was hypothesized that association with RNAP allows TopoI to rapidly relax negative DNA supercoils forming behind the elongating RNAP, thereby preventing the R-loops formation[28,30]. The chromosomal distribution of EcTopoI is currently unknown, although some sequence preferences have been reported in vitro[31,32]. Recently, genome-wide distribution of TopoI from *M. tuberculosis* (MtTopoI), *M. smegmatis* (MsTopoI), and *S. pneumoniae* (SpTopoI) was investigated using ChIP-Seq[29,33,34]. In all cases, the topoisomerase was shown to associate with actively transcribed genes, with particular enrichment upstream of RNAP peaks at promoter regions. These findings agree with the twin-domain model proposed by Liu and Wang[35,36], but do not necessarily imply direct TopoI-RNAP association.

Here, we map the EcTopoI-binding sites on the *E. coli* chromosome using ChIP-Seq. We demonstrate that during exponential growth, the enzyme is accumulated in regions with high levels of transcription, where it colocalizes with RNAP at promoters and transcription unit (TU) bodies. EcTopoI is also significantly enriched in extended, 12–15 kb-long, regions upstream of transcribed TUs. When transcription is inhibited by rifampicin (Rif), both EcTopoI and RNAP redistribute from TU bodies toward promoter regions, and EcTopoI disappears from the upstream regions. When a 14 kDa C-terminal domain of EcTopoI (14 kDa CTD) known to interact with RNAP[28] is overexpressed, EcTopoI enrichment in TU bodies and promoter regions decreases but remains unaffected in the upstream regions. By mapping the cleavage sites induced by an "intrinsically-poisoned"

EcTopoI mutant, we reveal that EcTopoI catalytic activity is increased in the upstream regions of highly-active TUs. Based on these data and pull-down experiments, we conclude that EcTopoI physically interacts with RNAP in TUs in a TopoI CTD-dependent manner. At the same time, and independently of RNAP, EcTopoI is attracted to extended upstream regions in front of TUs by negative supercoiling generated by transcription and removes supercoils. We finally demonstrate that prolonged overexpression of 14 kDa CTD or catalytically inactive EcTopoI Y319F mutant are lethal for cells and result in hypernegative supercoiling and accumulation of R-loops. We propose that the interaction between RNAP and EcTopoI is essential for DNA duplex restoration immediately upstream of the elongating RNAP, either by in situ relaxations of negative superhelicity or by structural clamping of DNA by the topoisomerase.

## Results

### EcTopoI is widely distributed over the *E. coli* genome, colocalized with RNAP, and enriched in regions with negative supercoiling

The topoisomerase I distribution along the *E. coli* chromosome was determined using ChIP-Seq with a DY330 strain derivative carrying a fusion of the *topA* gene with the SPA tag encoding sequence (Fig. 1, orange track). Three biological replicas were made, showing good reproducibility between them (Supplementary Fig. 2a, Pearson correlation >0.6). Using the MACS2 analysis pipeline[37], we detected 403 significantly enriched regions (e-value <0.001) present in all three replicas (Supplementary Figs. 2b–d). EcTopoI peaks tend to have a lower GC-content than the genome average (Supplementary Figs. 2e–j). Indeed, a positive correlation between peaks log-fold enrichment and the AT-content was observed (Spearmen correlation 0.36, *p* value 2.3e-5). Furthermore, the peaks appeared to be uniformly distributed over the entire chromosome. Of note, there was no enrichment of the EcTopoI signal at the terminator region of chromosome replication, in contrast to observations made for *M. smegmatis*[34].

We next determined whether the EcTopoI ChIP-Seq signal overlaps with the RNAP signal, a result that might be expected based on the published data about the interaction between the two enzymes[28]. We performed a ChIP-Seq experiment with a DY330 strain derivative expressing TAP-tagged RpoC (β′ RNAP subunit) (Fig. 1, green track). The RpoC ChIP-Seq signal correlated well with the published ChIP-Seq obtained for RpoB (β RNAP subunit; Spearman correlation 0.59, *p* value = 2.4e-158; Supplementary Fig. 3a)[38] and transcription level (RNA-Seq performed with exponentially growing *E. coli* DY330, Spearman correlation 0.55, *p* value 1.4e-133; Supplementary Fig. 3b). Overall, we found 3635 RpoC peaks with fold enrichment of at least 3, ~25% of which overlapped with earlier reported RpoB peaks (Monte-Carlo simulation with 10000 iterations, *p* value <1e-308; Supplementary Figs. 3c, 4a). 60% of topoisomerase peaks (243/403, Monte-Carlo simulation with 10000 iterations, *p* value = 4.9e-6; Supplementary Fig. 4b) overlapped with the RpoC peaks (Fig. 2a). Consistently, enrichment of RpoC is significantly higher within the EcTopoI-enriched regions compared to the outside regions (Welch *t*-test, *p* value <1e-308). Reciprocally, enrichment of EcTopoI is significantly higher inside the RpoC-occupied regions than outside of these regions (Welch *t*-test, *p* value <1e-308) (Fig. 2b). Colocalization of the RpoB and EcTopoI signals was also observed with a publicly available RpoB ChIP-Seq dataset for *E. coli* MG1655[38] (Supplementary Figs. 3d, e). Overall, we conclude that EcTopoI is significantly colocalized with RNAP on the *E. coli* chromosome in exponentially growing cells.

The ChIP-Seq signal of EcTopoI was also generally proportional to transcript abundance, with the highest enrichment values observed for 200 most highly-expressed transcription units (HETUs, expression level >31 FPKM) and, particularly, for rRNA operons (Supplementary

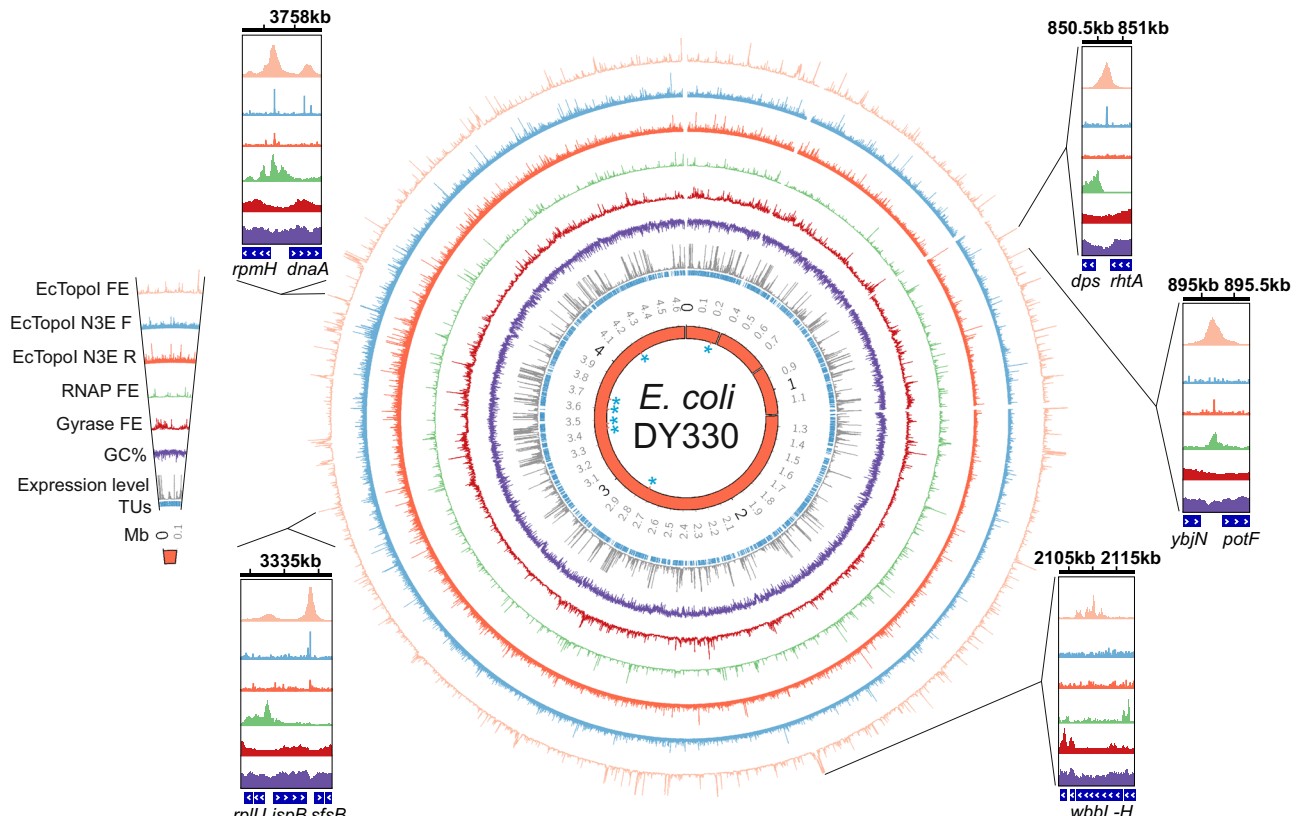

**Fig. 1 | Distribution of EcTopoI, RNAP, and gyrase enrichment peaks over the *E. coli* chromosome.** Circular maps show fold enrichment profiles of EcTopoI (ChIP-Seq, light orange; Topo-Seq, cyan and dark orange for two separate DNA strands), RNAP (ChIP-Seq, green), and DNA gyrase (Topo-Seq, dark-red). Additionally, GC-content (%, purple) and mean expression levels (FPKM, RNA-Seq, gray) for annotated TUs (inner blue segments) are also shown. Blue asterisks indicate positions of rRNA operons on the innermost orange ring representing *E. coli* DY330 genome.

The numbers on the outside of the orange ring indicate genome coordinates in megabase pairs (Mbs). Three gaps around ~0.3, ~0.8, and ~1.2 Mb correspond to deletions in the *E. coli* DY330 genome relative to the *E. coli* W3110 reference genome. Insets provide a zoom-in view of representative regions with high EcTopoI signals. Coordinates in kb are indicated on top of each inset. For ChIP-Seq, fold enrichment is given relative to the input sample in all figures. The maps were constructed with the Circos tool[95], and the insets were prepared using IGV[85].

Figs. 3f, g). In contrast, little or no EcTopoI enrichment was observed for 200 least-expressed TUs (LETUs, expression level <0.31 FPKM) (Fig. 2c).

Next, we analyzed the enrichment of EcTopoI and RNAP within TUs, and in upstream and downstream regions (Fig. 2c, d and Supplementary Fig. 3h). Metagene analysis indicated colocalization of EcTopoI and RNAP within the TU bodies, with the highest enrichment for both enzymes near the transcription start sites (TSS). A decreasing RNAP enrichment gradient toward the ends of TUs, presumably caused by premature transcriptional termination[39,40], was observed. A gradient with a similar slope was also detected for EcTopoI enrichment, suggesting that EcTopoI either directly follows elongating RNAPs or physically associates with the enzyme.

EcTopoI accumulated upstream but was depleted downstream of TUs, a result that is consistent with the predictions of the Liu & Wang twin-domain model that posits accumulation of negative supercoiling (a substrate of TopoI) upstream, i.e., behind the elongating RNAP[36]. Excessive accumulation of EcTopoI could be tracked up to 12–15 kb upstream of TSS for HETUs (Fig. 2c), suggesting that negative supercoiling diffuses over significant lengths of the *E. coli* chromosome. Interestingly, this range is significantly longer than that observed for eukaryotic chromatin, possibly due to the absence of supercoiling-"buffering" nucleosomes[41,42]. A small peak of EcTopoI and RNAP enrichment at TU ends may correspond to enrichment at promoter regions of closely packed adjacent genes or result from the physical association of the two enzymes at transcription termination sites.

Overall, our observations strongly support the association of EcTopoI with RNAP at TSSs and within the TUs, as well as with negatively supercoiled DNA upstream of actively transcribed genes.

## The RNAP inhibitor rifampicin causes EcTopoI re-localization to promoter regions

If EcTopoI interacts with RNAP, it should redistribute to promoters upon the treatment with rifampicin (Rif), an inhibitor that prevents RNAP escape into elongation[43,44]. In addition, if EcTopoI association with extended regions upstream of TUs is driven by excessive transcription-generated negative supercoiling, Rif treatment should abolish this association. To test these predictions, we performed EcTopoI ChIP-Seq in cells treated with the Rif prior to formaldehyde fixation. According to metagene analysis, EcTopoI enrichment along the lengths of HETUs bodies disappeared in Rif-treated samples, reaching values below the background (Fig. 2g). This was consistent with the disappearance of elongating RNAP from TU bodies in Rif-treated samples (Fig. 2h, k; ref. 44). Association of EcTopoI with upstream regions of HETUs was also abolished upon Rif treatment (Fig. 2g, j). Yet, the enrichment of EcTopoI at promoter regions of HETUs remained, although at lower levels compared to the untreated control (Fig. 2i, j). This decrease may be caused by the dissipation of transcription-induced negative supercoiling and/or by a more uniform redistribution of RNAP holoenzymes across promoters (since high-affinity promoters cannot be occupied by more than one RNAP molecule, remaining molecules become trapped by the Rif at weaker promoters). The latter scenario is supported by the observation that

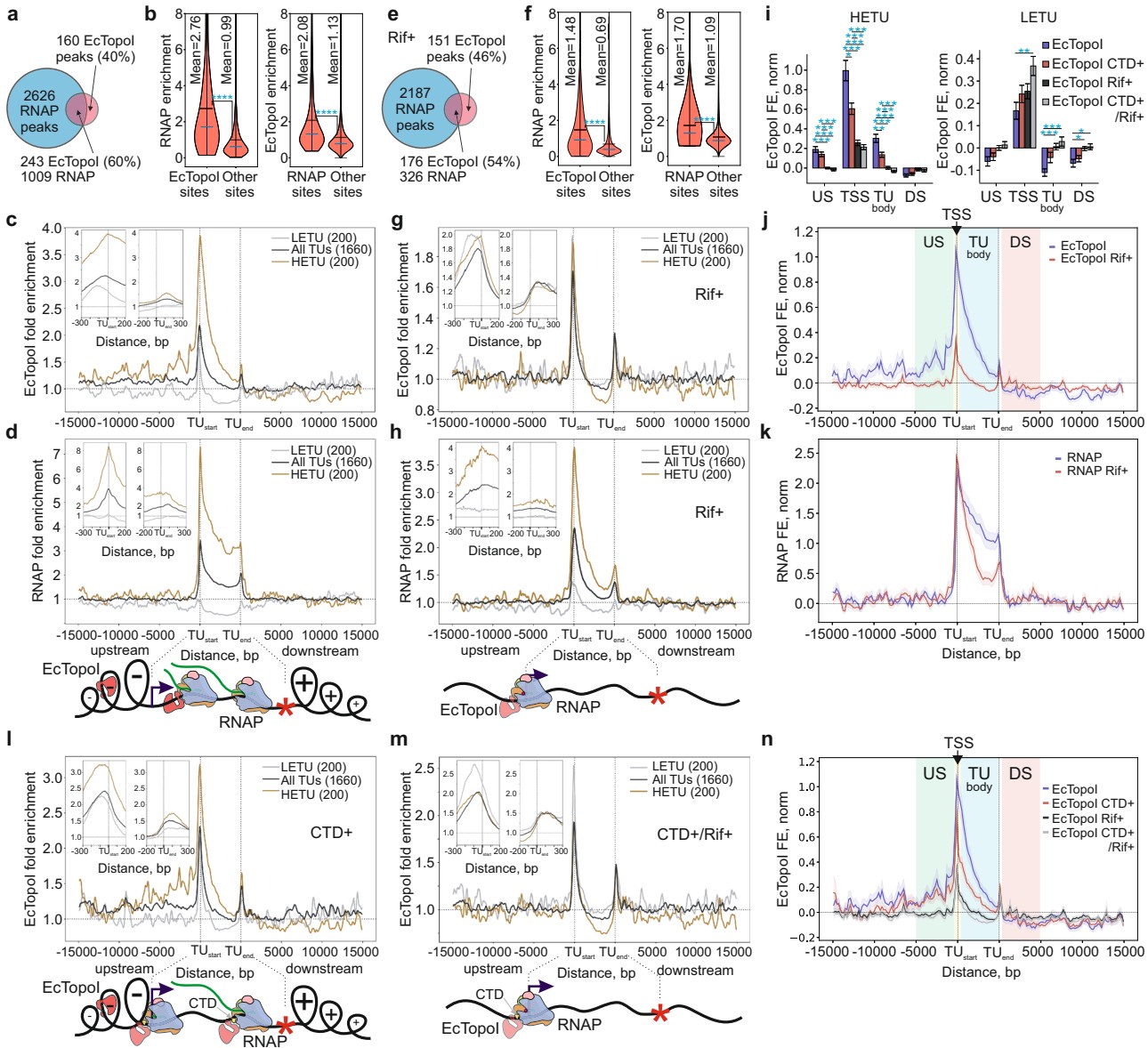

**Fig. 2 | EcTopoI is associated with RNAP and with regions of expected transcription-induced negative supercoiling. a** Venn diagram represents an overlap between the EcTopoI and RNAP peaks. **b** Violin plots of RNAP enrichment in EcTopoI peaks and outer regions (left), and EcTopoI enrichment in RNAP peaks and outer regions (right). The means and medians are indicated by black and blue lines, respectively. Statistically significant differences between means (two-sided *t*-test, *p* value = 8e-28 and 9e-123 for RNAP and EcTopoI enrichments, respectively) are indicated by asterisks. **c** Metagene plot of EcTopoI enrichment within TUs (middle), their upstream (left), and downstream (right) regions. Enrichment is shown for all TUs (black curve), highly-expressed (HETU, orange curve), and least-expressed (LETU, gray curve) sets. The number of TUs in each group is indicated in parentheses. The two insets show zoom-in views of EcTopoI enrichment near transcription start (TU start) and termination (TU end) sites. **d** Top, metagene plot of RNAP enrichment. Analysis and sets of TUs are the same as in **c**. Bottom, graphical representation of the Liu & Wang twin-domain model[36] showing localization of RNAP and EcTopoI according to the metagene plots in **c**, **d**. **e** Venn diagram represents an overlap between the EcTopoI and RNAP peaks (327 and 2513 peaks, respectively) in cells pretreated with Rif. **f** Violin plots of RNAP enrichment in EcTopoI peaks and outer regions (left), and of EcTopoI enrichment in RNAP peaks and outer regions (right) for Rif-treated cells (an RpoC ChIP-chip dataset from ref. 44). Means were compared by two-sided *t*-test. *P* values 9e-17 and 2e-6 for RNAP and EcTopoI enrichments, respectively, are indicated by asterisks. **g** Metagene plot of EcTopoI enrichment for cells pretreated with Rif. **h** Top, metagene plot of RNAP enrichment for cells pretreated with Rif. Bottom, graphical representation showing localization RNAP and EcTopoI according to metagene plots in **g**, **h**. **i** EcTopoI enrichment for HETUs (left) and LETUs (right) in untreated cells (blue bars), cells pretreated with Rif (black bars), and cells overexpressing 14 kDa CTD without Rif (red bars) or followed by Rif treatment (gray bars). Enrichment was quantified for normalized tracks in regions near the transcription start sites (TSS, ± 200 bp from transcription start site), 5 kb upstream regions (US), 5 kb downstream regions (DS), and TU bodies (TU). Enrichments were compared by a two-sided Welch *t*-test. *P* values <2e-3 are indicated by asterisks with the number of asterisks indicating the significance level, Bonferroni correction for multiple testing was applied. Bars represent mean values ± SEM, *n* = 200 TUs for all conditions. **j** Metagene plot of normalized EcTopoI enrichment for HETUs in cells untreated (blue curve) or pretreated with Rif (red curve). **k** Metagene plot of normalized RNAP enrichment for HETUs in cells untreated (blue curve) or pretreated with Rif (red curve). **l** Metagene plot of EcTopoI enrichment for cells overexpressing 14 kDa CTD (top) and a graphical representation of the localization of RNAP and EcTopoI (bottom). **m** Metagene plot of EcTopoI enrichment for cells overexpressing 14 kDa CTD followed by Rif treatment (top) and a graphical representation of the localization of RNAP and EcTopoI (bottom). **n** Metagene plot of normalized EcTopoI enrichment for HETUs in cells untreated (blue curve), or treated with Rif (black curve), and in cells overexpressing 14 kDa CTD without Rif (red curve), or followed by Rif treatment (gray curve). Confidence bands (±SEM) in panels **j**, **k**, and **n** are represented by light-colored profiles. Colored areas in panels **j** and **n** indicate regions used to quantify enrichment in panel **i** (US, TSS, TU body, and DS). For ChIP-Seq, fold enrichment is given relative to the input sample.

in Rif-treated samples, enrichment of both EcTopoI and RNAP is increased at LETU promoters (Fig. 2h, i). Be that as it may, EcTopoI and RNAP remained colocalized in Rif-treated cells (Welch t-test, p value <1e-308, Fig. 2e, f), sharing a significant number of enrichment peaks (Monte-Carlo simulation with 10,000 iterations, p value <1e-308; Supplementary Fig. 4c). Consistent with the re-localization of EcTopoI to promoters, EcTopoI peaks found by MACS2 in Rif-treated cells were narrower (median width 311 bp) and more AT-rich (43.5% GC) than peaks in untreated samples (Supplementary Fig. 2f). Overall, these results further support the EcTopoI interaction with elongating RNAP, promoter initiation complexes, and regions upstream of transcribed genes.

### Overexpression of EcTopoI 14 kDa CTD impairs interaction with RNAP

The 14 kDa CTD of EcTopoI interacts with *E. coli* RNAP in vitro[28]. We observed that RNAP copurified with both SPA-tagged EcTopoI (Supplementary Fig. 5a) and His-tagged 14-kDa-CTD (Supplementary Fig. 5b, c) during affinity chromatography from extracts of cells expressing corresponding tagged proteins. Overexpression of CTD but not of GFP control decreased the amount of RNAP copurified with SPA-tagged EcTopoI from *E. coli* DY330 cells (Supplementary Fig. 5d) (see Supplementary Methods for details). We, therefore, reasoned that overproduction of CTD might impair the RNAP:EcTopoI interaction in vivo and thus change the distribution of EcTopoI. Accordingly, we carried out EcTopoI ChIP-Seq experiments with cells overexpressing the EcTopoI CTD. To avoid possible biases caused by toxicity of prolonged overexpression of the CTD (Supplementary Fig. 5e, f), we induced cells with IPTG only for 1 h and then performed ChIP-Seq. As can be seen from Fig. 2n, overexpression of CTD indeed decreased EcTopoI enrichment in TUs and promoter regions (Fig. 2i). In contrast, EcTopoI enrichment in upstream regions of TUs in cells overproducing CTD and in control cells was similar (Fig. 2n, i). This enrichment was dependent on the level of transcription (Fig. 2l) and extended up to 12 kb for HETUs, implying that it was caused by transcription-induced negative supercoiling. Therefore, CTD does not compete with EcTopoI for the binding to negatively supercoiled DNA but impairs the RNAP:EcTopoI interaction. Treatment of CTD-expressing cells with the Rif resulted in EcTopoI enrichment profiles similar to those observed for Rif-treated control (compare Fig. 2m with Fig. 2g). Using ChIP-qPCR for two randomly chosen long (>2 kb) and medium-to-highly-transcribed TUs, we showed that CTD overexpression decreases the enrichment of RNAP toward the TUs ends (Supplementary Fig. 5g, h). We propose that this is caused by the premature stalling of transcription elongation complexes due to excessive negative supercoiling.

### EcTopoI is recruited to chromosomal regions with excessive negative supercoiling surrounded by topological barriers

Next, we examined EcTopoI distribution in 1529 *E. coli* intergenic regions (IRs, Supplementary Fig. 6a) in more detail. We observed significant EcTopoI enrichment at IRs flanked by highly-transcribed genes and/or having a high level of RNAP enrichment (Supplementary Fig. 6b, c). Irrespective of RNAP enrichment/transcriptional activity, high levels of EcTopoI enrichment were found at IRs that (i) were located between divergently transcribed genes (Supplementary Figs. 6d), (ii) harbored transcription factor-binding sites (Supplementary Fig. 6e, k), and (iii) were flanked by genes coding for membrane proteins (Supplementary Fig. 6f, g). Consistently, IRs that fulfilled all three criteria and were located between highly-transcribed genes had the highest level of EcTopoI enrichment (Fig. 3a and Supplementary Fig. 6h, i).

The meta-intergene analysis revealed that IRs flanked by divergent genes exhibit, on average, much higher EcTopoI signal than those located between convergent genes (Fig. 3c). Based on these results, we propose that EcTopoI is preferentially recruited to

regions with excessive negative supercoiling stabilized by local topological barriers (see representative examples in Supplementary Fig. 6j). These barriers may be generated by divergent transcription from highly complex promoters and by transertion process (a coupled transcription/translation/polypeptide chain translocation into the cell membrane)[45] (Fig. 3b).

### EcTopoI and DNA gyrase have mutually exclusive localization on the *E. coli* chromosome

EcTopoI and DNA gyrase have opposite binding preferences and activities: while EcTopoI is attracted to and relaxes negative supercoils, DNA gyrase is attracted to and removes positive supercoils[8,46–48]. A comparison of ChIP-Seq data for EcTopoI and Topo-Seq data for DNA gyrase[47] directly demonstrates that in vivo gyrase enrichment is significantly lower in regions occupied by EcTopoI and vice versa (Welch t-test, p value <1e-308, Fig. 3e). While EcTopoI is enriched upstream of HETUs (where transcription-induced negative supercoiling should be high) and depleted in downstream regions (where positive supercoiling should be accumulated), the gyrase enrichment shows the opposite pattern (Fig. 3d). We used Psora-Seq and GapR-Seq data available for *E. coli* to localize enrichment of topoisomerases with, respectively, regions of negative and positive supercoiling genome-wide. A signal of negative supercoiling revealed by Psora-Seq[49] matches the enrichment of EcTopoI, indicating that EcTopoI accumulation upstream of active TUs indeed colocalizes with increased negative supercoiling. Concordantly, a signal of positive supercoiling revealed by GapR-Seq[50] matches the enrichment of DNA gyrase in regions downstream of active TUs (Fig. 3d). Both gyrase enrichment downstream of TUs and EcTopoI enrichment upstream of TUs positively correlate with transcription activity and are abolished by Rif (Supplementary Fig. 6m). Furthermore, while EcTopoI is particularly enriched in IRs flanked by divergent genes (see above) where cumulative negative supercoiling is expected, the DNA gyrase signal is the highest for IRs between convergent genes (where cumulative positive supercoiling is expected) (Fig. 3c), in line with observations made in *M. tuberculosis*[33]. Together, these data indicate that EcTopoI and gyrase have opposing patterns of distribution genome-wide, fully consistent with the predictions of the Liu & Wang model[36].

### EcTopoI-induced DNA cleavage is increased in regions upstream of active TUs and decreased in TU bodies

The DNA topoisomerases binding and cleavage sites may not completely overlap[51,52]. To identify EcTopoI cleavage sites (TCSs) in vivo genome-wide, we constructed EcTopoI G116S/M320V, an "intrinsically-poisoned" double-mutant that forms stable covalent complexes with DNA[53]. As expected, continuous production of EcTopoI G116S/M320V from a plasmid led to growth inhibition (Supplementary Fig. 7a) and SOS-response (Supplementary Fig. 7b). EcTopoI G116S/M320V was transiently (30 min) expressed in *E. coli* DY330, and the trapped protein–DNA cleavage complexes were purified through a C-terminal StrepII tag fused with the mutant topoisomerase (Supplementary Fig. 7c). Expression of EcTopoI G116S/M320V had no apparent adverse effect on cell culture growth in the course of the experiment (Supplementary Fig. 7a). Topoisomerase-associated DNA fragments were isolated and subjected to strand-specific sequencing of ssDNA using the Accel NGS 1S kit, and the reads were mapped to the reference genome. Hereafter, we refer to this experimental pipeline as "Topo-Seq". The number of 3′-ends (N3E) was counted for every genomic position strand-specifically. Since EcTopoI forms a covalent intermediate with the 5′-end of a single-stranded break, it introduces and leaves the 3′-end unmodified, an increase in the N3E should mark a TCS. A total of 262 TCSs were identified in the *E. coli* genome (125 on the forward and 137 on the reverse strand). The TCSs determined by Topo-Seq, which identifies the sites of EcTopoI activity with single-base precision, significantly overlap with EcTopoI peaks detected by

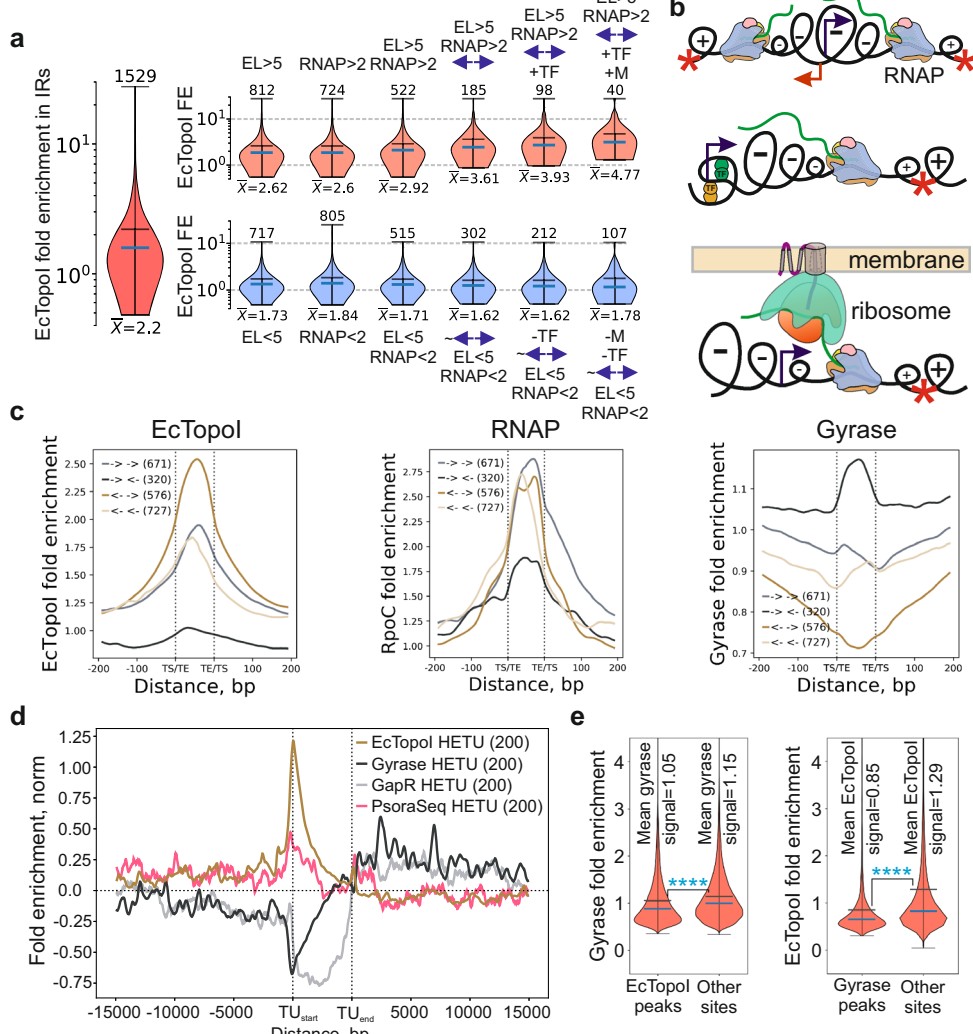

**Fig. 3 | IR features associated with the increased EcTopoI enrichment and mutual exclusion of EcTopoI and DNA gyrase genome-wide. a** Cumulative effect of transcription and local topological barriers on EcTopoI fold enrichment (FE) in IRs for CTD-/Rif- condition. Violin plots show the contribution of positive (top) and negative (bottom) factors on EcTopoI enrichment in IRs, including the expression level of adjacent genes (EL, FPKM units), FE of RNAP (RNAP, FE units), the orientation of adjacent genes (←→ for divergent and -←→ for not divergent orientation), annotated sites for transcription factors (+TF for at least one site is annotated and -TF if no sites are annotated), and membrane localization of proteins encoded by adjacent genes (+M for at least one gene encodes a membrane protein and -M if no such genes). The leftmost plot shows an overall EcTopoI fold enrichment in 1529 IRs. Means (numeric values are shown below) and medians are shown with horizontal black and blue lines, respectively; vertical axes are log-scaled. **b** Graphical representation of topological borders which trap negative supercoiling: divergent genes, complexly organized promoters, transertion. **c** Meta-intergene plots of EcTopoI (left), RNAP (center), and gyrase (right) enrichments in IRs for CTD-/Rif- condition. IRs were classified according to the orientation of flanking genes. The number of regions comprising a group is indicated in parentheses. **d** Metagene plot of EcTopoI ChIP-Seq (CTD-/Rif- condition), gyrase Topo-Seq (experiments with ciprofloxacin[47]), GapR-Seq[50], and Psora-Seq[49] enrichments. Analysis performed for the HETU set of TUs. **e** Violin plots of gyrase enrichment in EcTopoI peaks and outer regions (left) and of EcTopoI enrichment in gyrase peaks and outer regions (right). The mean and median are indicated by black and blue lines, respectively. Enrichments were compared by two-sided Welch t-test (p value = 5e-13 and 2e-198 for Gyrase and EcTopoI enrichments, respectively). Significance is indicated by asterisks. For ChIP-Seq, fold enrichment is given relative to the input sample.

ChIP-Seq (Fig. 4a) (Monte-Carlo simulation with 10,000 iterations, p value 3.5e-13; Supplementary Fig. 7d, e). Interestingly, several chromosomal regions with increased EcTopoI-binding (as evidenced by ChIP-Seq and ChIP-qPCR data shown in Supplementary Fig. 7h) and enhanced cleavage (as evidenced by Topo-Seq) also demonstrated high affinity to purified EcTopoI in vitro, revealing sequence specificity of the enzyme (Fig. 4b, and see below).

The DNA-binding and cleavage activities of EcTopoI were compared by metagene analysis. Both signals were increased over the background upstream of active TUs, where the GapR-Seq signal is significantly depleted, indicating the attraction of EcTopoI by transcription-generated negative supercoiling followed by relaxation of bound DNA (Figs. 4c, d, 3d for GapR-Seq data). Intriguingly,

compared to regions upstream of TUs, the cleavage activity of EcTopoI was significantly lower at promoters and within the TU bodies (Fig. 4d), i.e., at sites where the formation of complexes with RNAP is expected. Indeed, the overlap of TCSs with RNAP peaks is decreased (Monte-Carlo simulation with 10,000 iterations, p value 1.6e-2, Supplementary Fig. 7d, f). A similar pattern was reported for human TOP1 and RNAPII[51], implying that topoisomerase activity may be negatively regulated within the RNAP complexes both in prokaryotes and eukaryotes. Interestingly, while the cleavage is decreased in HETU bodies, it is increased inside LETUs, particularly towards their ends (Fig. 4e and Supplementary Fig. 7g). We speculate that in HETUs, where EcTopoI remains inactive, RNAP molecules move in convoys and mutually annihilate positive and negative supercoils[54,55]. In contrast, EcTopoI

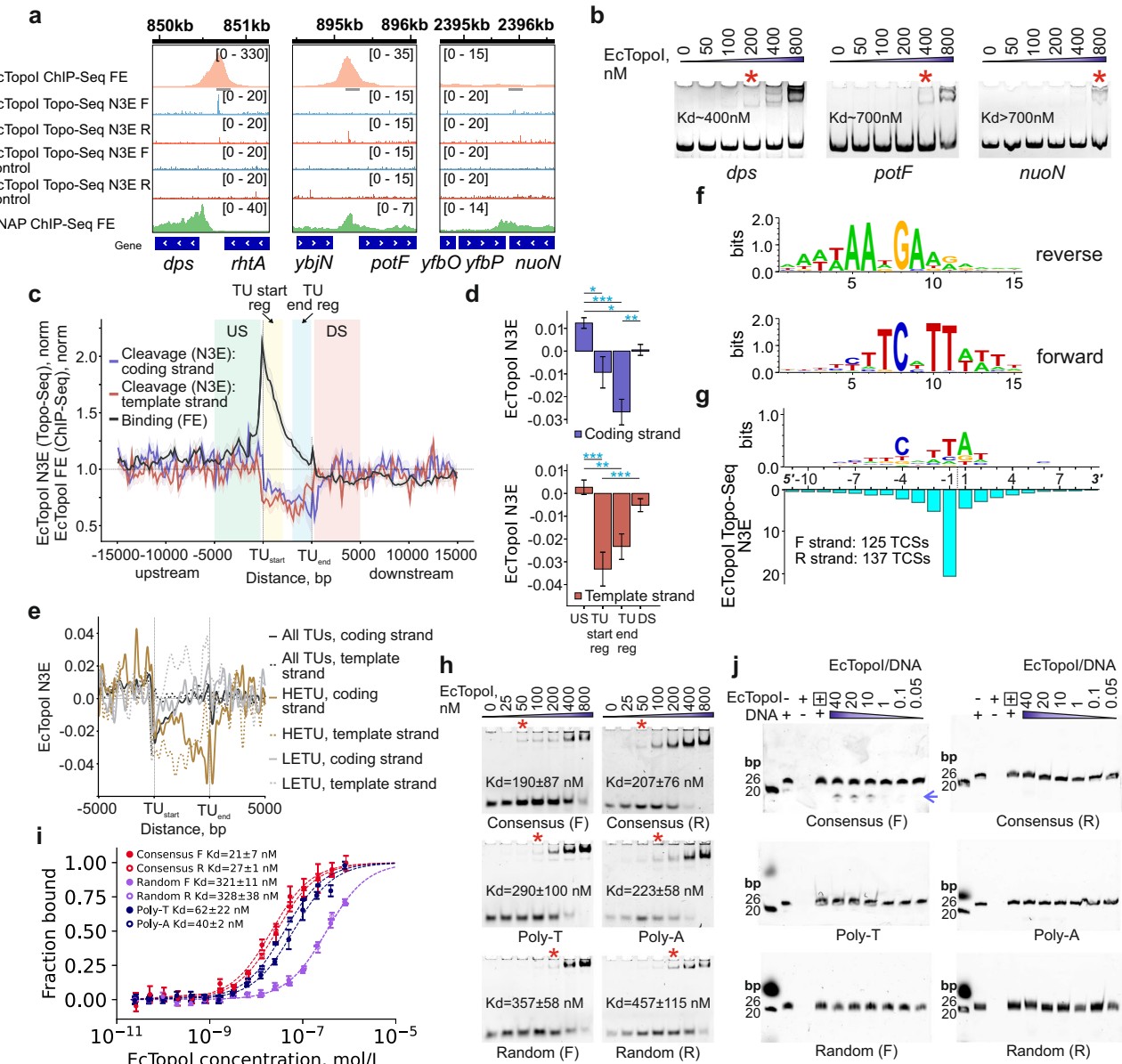

**Fig. 4 | Topoisomerase I cleavage sites (TCSs) identified by Topo-Seq and sequence specificity of binding and cleavage. a** Representative regions of the *E. coli* chromosome with EcTopoI ChIP-Seq peaks matching the EcTopoI Topo-Seq TCSs (*dps, potF*) and a region lacking EcTopoI-binding and activity (*nuoN*). EcTopoI cleavage activity is shown strand-specifically. A control, non-induced culture, is shown. Positions of regions used for ChIP-qPCR and affinity measurements are indicated by gray rectangles. **b** Affinity of purified EcTopoI to three amplified genomic regions from the panel **a** measured by EMSA. Red asterisks mark the lowest concentration of EcTopoI at which a gel-shift was detected. **c** Metagene plot of EcTopoI ChIP-Seq enrichment (untreated condition, black curve) and EcTopoI Topo-Seq cleavage signal (blue and red curves for coding and template strands, respectively). Confidence bands around the mean metagene signal are represented by ±SEM. Analysis was performed for the HETU set. Regions used for further quantification of enrichment in panel **d** are shown by colored areas on the plot. **d** Mean EcTopoI cleavage signal in different regions relative to HETUs. Means were compared by a two-sided Welch *t*-test. *P* values <4e-3 are indicated by asterisks

and Bonferroni correction for multiple testing was applied. Bars represent mean values ±SEM, *n* = 200 TUs. **e** Metagene plot of EcTopoI Topo-Seq cleavage activity for all TUs, LETU, and HETU sets. Cleavage is shown strand-specifically. **f** Logo of EcTopoI-binding motif identified in sequences under the ChIP-Seq peaks. A motif is shown in both orientations. **g** EcTopoI cleavage motif identified by alignment of TCSs. The cleavage site between nucleotides −1 and 1 is indicated by a dashed line. The cleavage signal (N3E) is plotted below. **h** Affinity of purified EcTopoI to oligonucleotides measured with EMSA. The binding of forward (left) and reverse-complement (right) oligonucleotides is shown. Red asterisks mark the lowest concentration of EcTopoI at which a gel-shift was detected (Supplementary Table 3). **i** Affinity of purified EcTopoI to oligonucleotides measured with MST. Data were represented as mean values ± SEM, minimum of three independent MST experiments were performed. For source data, see Table S1 in the Source Data file. **j** Cleavage of oligonucleotides by purified EcTopoI. A control with EcTopoI inactivated by a high temperature is indicated with a boxed + sign. Cleavage products are marked with a blue arrow. For ChIP-Seq, fold enrichment is given relative to the input sample.

activity is needed to remove excessive negative supercoiling generated by individual RNAP molecules in LETUs.

## Identification of the EcTopoI-binding and cleavage motif

We used ChIPMunk[56] to find overrepresented motifs in EcTopoI ChIP-Seq peaks sequences. A strong motif was detected in more than 90%

of enrichment peaks for all conditions tested (Supplementary Fig. 7i, j). The motif was AT-rich, strongly asymmetric, with a conserved central TCNTTA/T part (Fig. 4f) and limited similarity to known DNA motifs in *E. coli* (Supplementary Fig. 7k). Initially, we tested the single-stranded DNA oligonucleotides for their ability to bind EcTopoI in vitro using gel-based EMSA. Since the putative EcTopoI-binding

motif is asymmetric, oligonucleotides corresponding to both strands of the consensus motif ("F"-forward, "R"- reverse) were used. Poly-T, Poly-A, and two complementary random-sequence oligonucleotides of equivalent length served as controls. As can be seen from Fig. 4h, both consensus oligos bound EcTopoI with comparable affinities (apparent Kd ~190–207 nM). In control, the binding affinities for random oligos were noticeably lower (Kd ~350–450 nM), whereas Poly-T and Poly-A showed intermediate affinities (Kd ~223–290 nM) (see Supplementary Table 3). The competition-binding EMSA revealed that the EcTopoI complex with consensus oligos was refractory to the action of control oligos (up to 16-fold molar excess, except for the consensus R and Poly-A oligos competition). In contrast, complexes with control oligos were highly susceptible to challenge by consensus oligos (50–90% dissociation at 1:1 molar ratio) (Supplementary Fig. 7l). These results indicate stronger and more stable TopoI binding to consensus DNA than to random oligos. Yet, Consensus R oligo appeared to bind EcTopoI as efficiently as Poly-A DNA in competition experiments, which corresponds to $K_D$ values reported above (Supplementary Fig. 7l).

To determine the EcTopoI-DNA-binding affinities more accurately, we used microscale thermophoresis (MST). Consistent with EMSA data, consensus oligos exhibited much higher binding affinities ($K_D$~21–27 nM) than random oligos ($K_D$~321–328 nM), while Poly-T and Poly-A oligos demonstrated intermediate affinities ($K_D$ ~40–62 nM) (Supplementary Fig. 4i and Supplementary Table 3). Overall, the binding affinities for EcTopoI revealed in our experiments rank as Consensus F > Consensus R-Poly-A > Poly-T≫random oligos.

Next, to test whether EcTopoI has a specific cleavage motif, we aligned sequences around the established TCSs. The identified cleavage motif was very similar to the binding motif identified using ChIP-Seq. As shown in Fig. 4g, EcTopoI preferentially cleaves a TA dinucleotide located 4 nt downstream of the conserved position of C. The cleavage motif was further validated in vitro by DNA-cleavage assay using the same consensus and control oligos as in EMSA. The cleavage was only observed for Consensus F oligo with a sequence that matches the deduced cleavage motif (Fig. 4j). Overall, we conclude that while EcTopoI prefers to bind an AT-rich single-stranded DNA, the presence of a single C residue within an AT-rich patch is necessary for efficient cleavage. Earlier in vitro experiments demonstrated that type-IA topoisomerases, including EcTopoI, specifically recognize a C residue and cleave DNA 4 nt downstream[32,57,58]. Our results extend these observations and show that a C in a specific context is required for EcTopoI cleavage in vivo.

## Impairing the RNAP:EcTopoI interaction mimics inactivation of EcTopoI and is deleterious to cell growth

If the RNAP:EcTopoI complex has a physiological role, uncoupling the RNAP:EcTopoI interaction shall have an impact on cell viability. Indeed, overnight overexpression of the 14 kDa EcTopoI CTD dramatically inhibited colony formation; 1-hour overexpression had a bacteriostatic effect and led to a ~2-fold decrease in the number of CFUs (Supplementary Fig. 5e, f). Consistently, induction of CTD overexpression slowed culture growth ~75 min post-induction (Fig. 5aiii). Overexpression of GFP or full-length EcTopoI had no such effect (Figs. 5ai, ii). Cells expressing the CTD formed filaments (Supplementary Fig. 5b) and underwent SOS-response, indicating accumulation of DNA breaks (Supplementary Fig. 8a). In order to further characterize the role of topoisomerase activity within the RNAP:EcTopoI complex, we overexpressed full-length catalytically inactive mutant EcTopoI Y319F from the pCA24 plasmid. Overexpression of this mutant was extremely toxic, presumably indicating the substitution of wild-type, chromosomally-encoded EcTopoI with the plasmid-encoded mutant in the complex (Supplementary Fig. 8b).

As an alternative strategy to disrupt the RNAP:EcTopoI interaction, we constructed *E. coli* strains with *topAΔ11* (recapitulates the well-known *topA66* mutation[59]), *topAΔ14*, and *topAΔ30* mutations. These mutations lead to the production of EcTopoI lacking an 11 kDa portion of CTD, the entire 14 kDa CTD, or a longer 30 kDa fragment that includes both the CTD and the Zn-binding domain, respectively (Fig. 5c). Since EcTopoI interacts with RNAP through both the CTD and the Zn-binding domain[28], we expected that *topAΔ11* and *topAΔ14* would decrease the interaction with RNAP, while *topAΔ30* will abolish it. In fact, the *topAΔ30* deletion inactivated EcTopoI[60,61]. While *topAΔ11* and *topAΔ14* mutant strains formed colonies indistinguishable from wild-type, colonies formed by the *topAΔ30* mutant were heterogeneous: most were much smaller than wild-type, and others had a wild-type appearance. Whole-genome sequencing revealed that cells from *topAΔ11* and *topAΔ14* colonies had no additional mutations, while cells from nearly all fast-growing *topAΔ30* colonies harbored mutations in the gyrase genes (Fig. 5f). Amplification of the chromosomal region containing the *parC* and *parE* genes encoding topoisomerase TopoIV was detected in one of the *topAΔ30* clones (#17). Two *topAΔ30* clones (##7 and 9) had no additional mutations. Clone #7 was used for further analysis. The growth curve analysis showed that while the doubling times of *topAΔ11* and *topAΔ14* strains were indistinguishable from that of the wild-type (30 min), the *topAΔ30* clone #7 grew slower, with a doubling time of 36 min (Fig. 5d). The fraction of long (>2* mean length of wild-type cells) cells was considerably higher in *topAΔ11* and *topAΔ14* cultures (11%) compared to the wild-type (0.3%). The fraction of long cells was 23% in *topAΔ30* culture, and these cells were much longer than the wild-type or other mutant cells (Fig. 5e). Consistent with the increased frequency of longer cells, the *topAΔ11* and *topAΔ14* mutants were outperformed by the wild-type in long-term competition experiments (Supplementary Fig. 8c). Overall, we conclude that EcTopoI catalytic activity within the RNAP:EcTopoI complex is required for optimal growth since its absence phenotypically mimics both the the the *topAΔ30* mutation and the deletion of the entire *topA* gene. Suppressor mutations observed in *topAΔ30* clones might reduce global negative supercoiling by the gyrase, thus compensating for the deleterious effects of EcTopoI inactivation.

## Impairing the RNAP:EcTopoI interaction leads to excessive negative supercoiling and accumulation of R-loops

It has been suggested that the interaction of EcTopoI with RNAP may prevent the formation of R-loops behind the elongating transcription complex, thus helping restore the DNA duplex and increasing the processivity of transcription[30]. To test this idea, we examined changes in the topological state of plasmids in cells overexpressing the 14 kDa CTD, a condition that impairs the RNAP:EcTopoI interaction (above). If uncoupling leads to the accumulation of R-loops, hypernegative supercoiling of plasmid DNA shall be expected[16]. Indeed, in agreement with earlier reported data of ref. 28, negative supercoiling of plasmids increased dramatically in cells overexpressing the CTD. In fact, over time, a hypercompacted plasmid form appeared in these cells. This form migrated faster than any other topoisomer (Fig. 5giii) during electrophoresis in the presence of chloroquine and may have corresponded to plasmids containing R-loops[16]. Correspondingly, overexpression of catalytically inactive EcTopoI Y319F led to an even more dramatic and rapid accumulation of hypercompacted plasmids (Supplementary Fig. 8d). No excessive supercoiling or plasmid hypercompaction was observed upon overexpression of GFP or full-length EcTopoI (Figs. 5gi, ii); the latter condition led to plasmid relaxation, as expected.

Another condition that affects RNAP:EcTopoI interaction is the deletion of the C-terminal region of EcTopoI (see above). To assess the level of plasmid supercoiling in BW25113 *topA* mutants and the wild-type strain, plasmids were extracted from overnight cultures, and topoisomers were analyzed by electrophoresis. Plasmids purified from *topAΔ11* and *topAΔ14* clones, where RNAP:EcTopoI interaction is

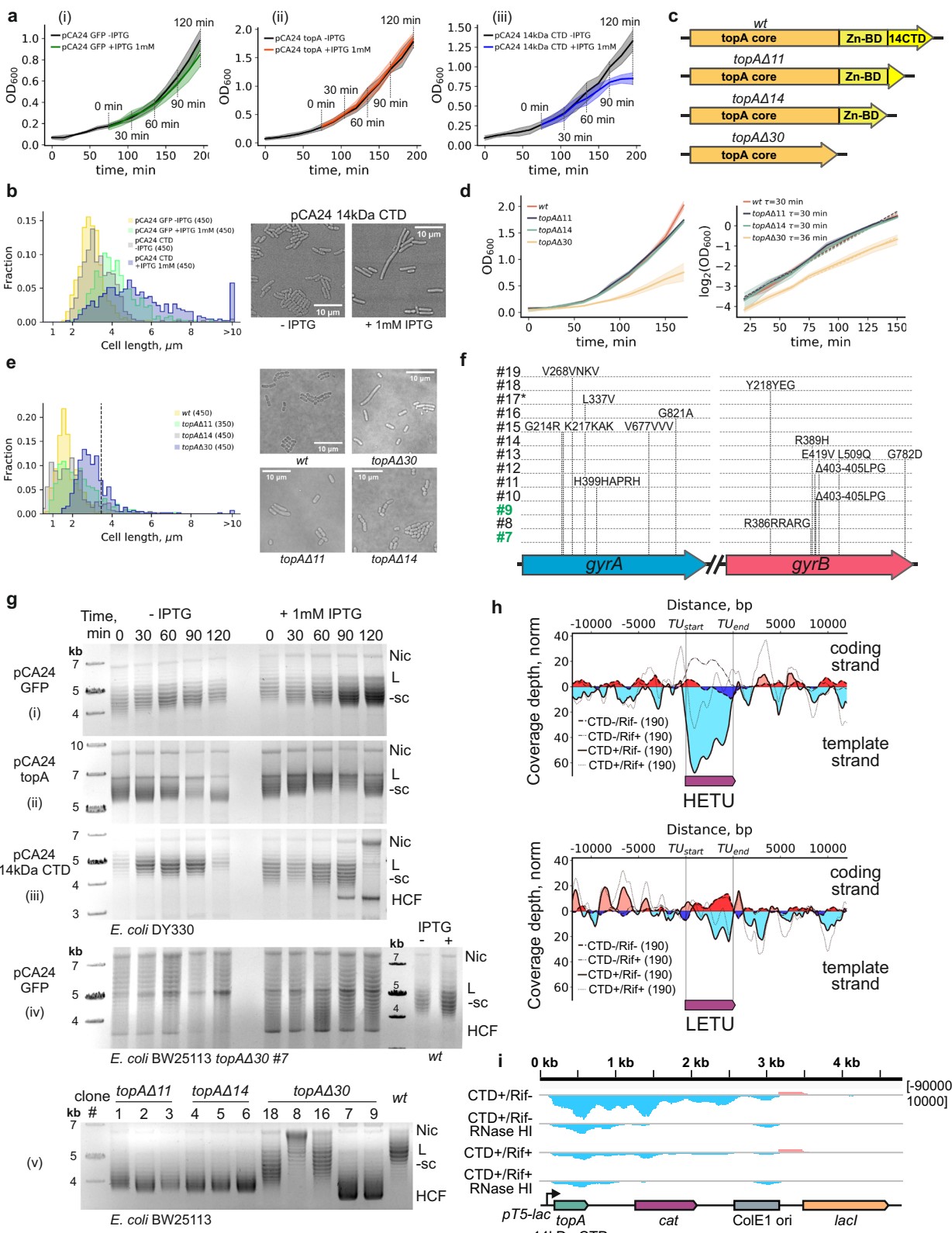

reduced, had higher levels of negative supercoiling than plasmids purified from the wild-type control (Supplementary Fig. 5gv). Supercoiling levels varied dramatically for plasmids purified from different *topAΔ30* clones, where EcTopoI is inactivated and there is no interaction with RNAP. The highest level of negative supercoiling, approaching that of hypernegatively supercoiled plasmids from CTD-expressing cells, was observed for plasmids purified from clones ##7

and 9 that lacked suppressing mutations (Supplementary Fig. 5gv). Plasmid hypercompaction in *topAΔ30* clone #7 was more prominent when expression of plasmid-borne *gfp* was induced with IPTG (Fig. 5giv), indicating that uncoupling of RNAP:EcTopoI complex or EcTopoI inactivation and active transcription together contribute to this phenomenon. These results are consistent with the R-loop accumulation hypothesis.

**Fig. 5 | Uncoupling of RNAP:EcTopoI complex is toxic for cells, leads to hypernegative supercoiling of plasmids, and accumulation of R-loops.**
**a** Growth curves for *E. coli* DY330 *topA*-SPA harboring pCA24 GFP (i), pCA24 14 kDa CTD (ii), or pCA24 topA (iii) plasmids. Data for induced (+IPTG 1 mM) and non-induced (−IPTG) cultures are shown. Shade represents a 0.95 confidential interval of the mean of three biological replicates. Gray lines mark aliquots collection for plasmid extraction. **b** Quantification of cell length in CTD or GFP producing cultures (left). Cells >10 μm are collected into an overflowing bin. The number of cells is indicated in parentheses. Representative fields are shown on the right.
**c** Graphical representation of truncated versions of a *topA* gene constructed by recombineering in *E. coli* BW25113. **d** Growth curves of *E. coli* BW25113 strains with truncated versions of *topA* and the wild-type control (left). Shade represents 0.95 confidential intervals of the mean. Quantification of doubling time for exponential regions of growth curves (right). **e** Quantification of cell length for *E. coli* BW25113 strains with truncated versions of *topA* and the wild-type (left). Vertical dashed line marks 2*mean cell length for wild-type. Representative fields are shown on the right. **f** Mutations in gyrase genes (*gyrA*, *gyrB*) found in *E. coli* BW25113 *topAΔ30* clones. An asterisk indicates amplification of a chromosomal region containing TopoIV genes; clones lacking compensatory mutations are highlighted in green. **g** Supercoiling of pCA24 GFP (i), pCA24 topA (ii), and pCA24 14 kDa CTD

(iii) plasmids extracted from exponentially growing *E. coli* DY330 *topA*-SPA. Time-points correspond to panel **a**. Supercoiling of pCA24 GFP (iv) plasmid extracted from exponentially growing *E. coli* BW25113 *topAΔ30* (time-course, on the left) or *E. coli* BW25113 *wt* (two rightmost lanes). (v) Supercoiling level of the pCA24 GFP plasmid extracted from overnight cultures of different clones of *E. coli* BW25113 *topA* mutants and from the wild-type control. Clone numbers correspond to panel **f**. Nic - nicked plasmid, L - linear plasmid, −sc - negatively supercoiled plasmid, HCF - hypercompacted plasmid. **h** Metagene plots of normalized strand-specific read coverage depth obtained in DRIP-Seq experiments for *E. coli* DY330 *topA*-SPA for HETU (upper panel, rRNA operons were excluded) and LETU (lower panel) sets. Schematic TUs are shown below. Data for CTD-/Rif- condition are shown with a dashed line, coverage depths for the coding and template strands are indicated by dark-red and dark-blue fillings, respectively. Data for CTD+/Rif− condition are shown with a solid line, coverage depths for the coding, and template strands are indicated by light-red and light-blue fillings, respectively. **i** DRIP-Seq data for pCA24 14 kDa CTD for CTD+/Rif− and CTD+/Rif+ conditions and corresponding RNase HI-treated controls. Coverage depths for "−" and "+" strands are shown in light-blue and light-red, respectively. A linearized map of the plasmid is shown below. Source data are provided as a Source Data file.

To directly observe the accumulation of R-loops, we performed strand-specific DRIP-Seq at conditions identical to those used for ChIP-Seq. R-loops accumulation in HETUs was revealed upon over-expression of the 14 kDa CTD (Fig. 5h). In addition, a large portion of the pCA24 14 kDa CTD expression plasmid, including the *topA* fragment encoding the 14 kDa CTD and the *cat* antibiotic resistance gene transcribed in the same direction, was covered by R-loops upon induction of transcription. Rifampicin dramatically reduced R-loops accumulation (Fig. 5i). Dot-blot analysis also demonstrated an increased level of R-loops formation in response to CTD or EcTopoI Y319F overexpression (Supplementary Figs. 8e, f). We conclude that a complex between catalytically active EcTopoI and RNAP is required to prevent R-loop accumulation. Transcription-induced R-loops that accumulate upon uncoupling of the RNAP:EcTopoI association may be the cause of toxicity observed in the absence of topoisomerase/its activity or when the complex is disrupted by a competitor.

## Discussion

In this study, several important observations are made. First, we show that an interaction between EcTopoI and RNAP is required for *E. coli* cell viability. Disruption of such interaction leads to hypernegative DNA supercoiling and dramatic R-loops accumulation. Our data provide a mechanistic explanation for RNAP:EcTopoI complex function and show that EcTopoI is required for R-loops formation control. Second, we demonstrate directly that EcTopoI and DNA gyrase are localized in extended upstream and downstream regions of TUs, respectively, illustrating the diffusion of unconstrained supercoils generated by transcription in accordance with the twin-domain model. Finally, we revealed that both DNA topology and local sequence patterns define the localization and activity of EcTopoI genome-wide.

In vitro, bacterial topoisomerase I efficiently relax negatively supercoiled DNA[8,48]. Therefore, in vivo, the enzyme should relieve negative supercoiling generated by transcription and, possibly, replication and balance the DNA gyrase activity to maintain a physiological level of supercoiling[62,63]. *E. coli* topoisomerase I was demonstrated to interact through its Zn-binding and C-terminal domains with the RNAP β' subunit[28,64]. Eukaryotic TOP1 and RNAPII, as well as mycobacterial and streptococcal TopoI and RNAP, were also shown to interact[29,51,65]. These interactions must have evolved independently since eukaryotic TOP1, and prokaryotic TopoI are evolutionarily distant from each other[66], while MtbTopoI appears to interact with MtbRNAP through a domain other than the one used by *E. coli* enzyme[67]. Yet, the ubiquitous presence of this interaction suggests the existence of common topological problems associated with transcription that need to be resolved in all domains of life[51,65].

## Genome-wide localization patterns of EcTopoI and DNA gyrase support the twin-domain model

Considering the fact that EcTopoI interacts with RNAP and has an increased affinity to negatively supercoiled DNA, it should be particularly enriched at highly-transcribed regions of the bacterial chromosome. In full agreement with this expectation, we observed that the ChIP-Seq signal from EcTopoI is enriched in the bodies and promoters of active TUs. We also detected prominent EcTopoI enrichment in extended, up to ~12–15 kb, regions upstream of active TUs. We propose that this enrichment defines the range that transcription-induced supercoiling can diffuse along the *E. coli* chromosome. Alternatively, the enrichment of EcTopoI in upstream regions might be mediated by transcription-induced bacterial "chromatin" remodeling, though we did not observe any significant skew in the enrichment of Fis, HNS, MatP, and MukB nucleoid-associated proteins in these regions (Supplementary Fig. 8g, h). Analysis of EcTopoI enrichment in IRs supports the notion that the enzyme is attracted to topologically constrained regions that accumulate negative super-coiling. In contrast, an inverted enrichment pattern is observed for DNA gyrase (by Topo-Seq) and GapR (by ChIP-Seq), proteins known to (i) act upon/interact with positively supercoiled DNA and (ii) avoid negatively supercoiled DNA. Taken together, these observations provide a whole-genome validation of the twin-domain model proposed by ref. 36.

## EcTopoI interacts with RNAP in vivo

The EcTopoI signal in TU bodies overlaps with the RNAP signal and is abolished by rifampicin. In contrast, EcTopoI enrichment, as well as RNAP enrichment at promoters, is unaffected by the Rif. These data further support a tight linkage of EcTopoI with RNAP. Furthermore, overexpression of EcTopoI CTD, which suppresses RNAP:EcTopoI interaction, decreases the enrichment of topoisomerase in TUs and promoters while leaving signals in extended upstream regions unaffected. Since CTD does not affect transcription elongation, the results support the existence of an RNAP:EcTopoI complex that is formed during transcription initiation and persists during transcription elongation in vivo. In contrast to *E. coli*, enrichment of streptococcal SpTopoI in both the upstream regions and promoters is abolished by the Rif, possibly indicating a distinct mechanism of RNAP:TopoI interaction[29].

## *E. coli* and *Mycobacterium* rely on different versions of the twin-domain model

Analysis of the genome-wide distribution of mycobacterial topoisomerases (DNA gyrase and TopoI) and RNAP reported by Nagaraja

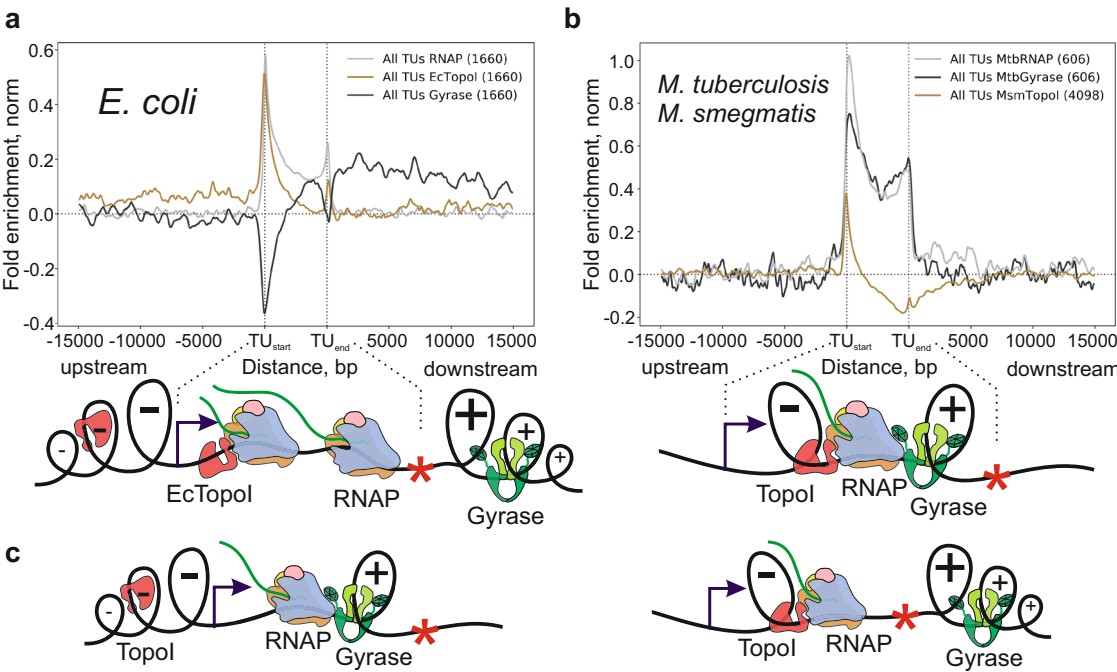

**Fig. 6 | Possible variations of the twin-domain model.** Average normalized enrichment of TopoI, DNA gyrase, and RNAP over transcription units of *E. coli* (**a**, "open" model) and *Mycobacterium* (**b**, "closed" model). Graphical representations of twin-domain sub-models are shown below. ChIP-Seq data for *M. tuberculosis* MtbRNAP, MtbGyrase, and *M. smegmatis* MsmTopoI was taken from publicly available datasets[33,34,96]. **c** Other "semi-open" hypothetical variations of the twin-domain model, based on the interaction of key topoisomerases (TopoI, DNA gyrase) with RNAP and their activity within a complex. For ChIP-Seq, fold enrichment is given relative to the input sample.

and co-workers showed colocalization of all three enzymes[33,34]. We re-examined the published ChIP-Seq data. Indeed, a significant colocalization between RNAP and gyrase, and RNAP and TopoI was observed (Supplementary Fig. 9). Surprisingly, in *Mycobacterium*, both topoisomerases are enriched within the TU bodies with the highest signal near the transcription start sites. In contrast to *E. coli*, there is no evidence for supercoiling diffusing away from the TUs (Fig. 6a, b). These patterns define two possible variations of the original Liu & Wang's scheme: an "open" model for *E. coli* (and, likely, *S. pneumoniae*, Supplementary Fig. 9k) in which supercoiling domains extend on DNA bi-directionally over a substantial distance from transcribing RNAP and a "closed" model for *Mycobacterium* where the supercoiling domains are trapped within RNAP-topoisomerase I/gyrase complex and cannot escape. Possibly, both mycobacterial topoisomerases form a relatively stable complex with RNAP, in which they fully relax the supercoils generated within TUs during transcription and, therefore, their activity is not needed in adjacent regions. Hypothetically, "semi-open" models may also exist: (i) when gyrase is highly-active in complex with RNAP while TopoI does not interact with RNAP, allowing negative supercoils to diffuse freely upstream of TUs; (ii) alternatively, when only TopoI is active within the RNAP complex, allowing positive supercoils to diffuse downstream of TUs where they are relaxed ahead of RNAP by free gyrase (Fig. 6c). Additional variations of these models may also be possible depending on the balance of topoisomerase and RNAP activities. If topoisomerase acts faster than the rate at which RNAPs generate supercoiling, diffusion of supercoiling will be limited due to rapid relaxation by topoisomerases. Following this logic, EcTopoI in complex with RNAP may allow a portion of unconstrained supercoiling to diffuse upstream, where it is subsequently relaxed by free topoisomerase. Our observation that EcTopoI does not actively cleave DNA in TUs when associated with RNAP (Fig. 4c, d, and see below) is consistent with this hypothesis.

## EcTopoI cleavage activity might be negatively regulated in complex with RNAP

By using a transient expression of an "intrinsically-poisoned" EcTopoI mutant, we performed Topo-Seq, an approach that allowed us to identify EcTopoI cleavage sites genome-wide. A similar approach was applied earlier for TopoI from *M. smegmatis* (MsmTopoI)[34]. Interestingly, despite the prominent binding of both EcTopoI and MsmTopoI to promoter regions, neither enzyme cleaves DNA there. However, the pattern of activity in TU bodies and regions upstream of TUs is different for the two enzymes. First, MsmTopoI remains active (i.e., cleaves DNA) in TUs bodies[34]. In contrast, EcTopoI is inactive at the beginning of TUs, but its activity increases toward the end of TUs, particularly in LETUs. This behavior may reflect activation of EcTopoI by extensive torsional stress generated by individual RNAP molecules expected in LETUs. Conversely, in HETUs, a caravan of RNAP molecules mutually annihilate positive and negative supercoils[54,55]. Second, no activity of MsmTopoI was detected in extended regions upstream of TUs, while EcTopoI activity in these regions is significantly increased, again illustrating the proposed "open" and "closed" variants of the twin-domain model. Overall, the data for EcTopoI resemble the activity pattern of eukaryotic TOP1, which remains inactive in complex with RNAPII until it is triggered by an RNAPII stalled by torsional stress[51]. A similar, independently evolved mechanism could be at work for bacterial topoisomerases. A naturally occurring modification of EcTopoI, $N^\varepsilon$-acetylation of lysins, was reported to reduce the enzyme's activity in vivo[68]. We speculate that this modification can regulate the activity of EcTopoI when it's in complex with RNAP. We predict that EcTopoI activity is inhibited by acetylation at promoters and at the beginning of TU bodies, and is activated by deacetylation (probably, by the CobB protein) at the end of TUs. This deacetylation may be triggered by conformational changes within the RNAP:EcTopoI complex caused by RNAP stalling or by extensive torsional stress. Alternatively, the apparent absence of the TCSs within active TUs may be explained by

the activity of transcription-coupled DNA repair pathways which might remove such complexes in situ[69,70].

## EcTopoI is not involved in chromosome decatenation in the Ter region

Topoisomerase I can catenate and decatenate ssDNA and dsDNA circles containing nicks in vitro[34,71,72]. Recently, an enrichment of TopoI activity was found in the Ter region of *M. smegmatis* chromosome[34]. This bacterium lacks classical decatenating topoisomerases, TopoIII and TopoIV, and, thus, TopoI is likely to be involved in chromosomal decatenation after replication. Despite the similarity in TopoI in vitro activities in both bacterial species, we did not observe any specific binding or cleavage by EcTopoI near Ter regions of the *E. coli* chromosome, indicating that this enzyme is not involved in chromosome decatenation.

## EcTopoI has sequence specificity in vivo

By combining ChIP-Seq and Topo-Seq, we identified the in vivo binding/cleavage motif of EcTopoI, which was validated in vitro. The EcTopoI-binding motif is asymmetric, AT-rich, and contains a single conserved C nucleotide. This C is located four nucleotides upstream of the cleavage site, which occurs at a TA dinucleotide. Our data indicate that AT-rich sequences are suitable binding substrates for EcTopoI, but the properly positioned C residue is strictly required for cleavage. These data are in agreement with previous in vitro observations made for different type-IA topoisomerases[57,58,73–75]. Likely, this requirement is characteristic of the entire protein family. For the optimal activity of EcTopoI, the binding/cleavage motif should be "activated" by melting of DNA, excessive negative supercoiling upstream of TUs, or, perhaps, by hypothetical signaling such as deacetylation of EcTopoI when in complex with RNAP at the end of TUs.

## RNAP:EcTopoI complex is required for R-loops formation control

Overexpression of EcTopoI competitors (14 kDa CTD or EcTopoI Y319F mutant) is toxic for *E. coli*[28]. We demonstrate that cells overexpressing the CTD form filaments (Fig. 5b) and accumulate R-loops genome-wide (Fig. 5h and Supplementary Fig. 8e). In addition, plasmids purified from these cells are hypernegatively supercoiled (Fig. 5g). Earlier, such changes were detected in *topA-null* mutants[14,15,76,77] as well as in this work for the *topAΔ30* mutant, which lacks the DNA relaxation activity[60,61,78]. It was previously proposed that the primary role of TopoI is to prevent extensive R-loops formation by relaxation of excessive negative supercoiling[13,16]. EcTopoI remains fully active upon CTD overexpression, as indicated by enrichment in upstream regions of active TUs (Fig. 2l). Thus, it appears that physical interaction between RNAP and EcTopoI is required to efficiently prevent R-loops formation during transcription.

Why then *E. coli topAΔ11* and *topAΔ14* mutants have close to wild-type viability (despite slightly increased cell length, more negatively supercoiled DNA, and decreased fitness in competition experiments)? We propose that the Zn-binding domain of EcTopoI is primarily responsible for the binding to RNAP, and its affinity to RNAP is sufficient for complex formation. Of note, molecular dynamics simulation predicted several residues of the Zn-binding domain, but not CTD, to be involved in the interaction with RNAP[64]. We assume that when CTD is overexpressed and binds to RNAP, it sterically excludes topoisomerase from the complex.

Except for rapidly acquired mutations in the gyrase and TopoIV genes[9,10,13], we did not observe any specific changes in expression levels of DNA topology-related genes (*topB, parC, parE, rnhA, rnhB, gyrA, gyrB*) upon uncoupling the RNAP:EcTopoI complex by EcTopoI Y319F mutant overexpression (Supplementary Fig. 8i). Therefore, the RNAP:EcTopoI interface may be a promising target for developing a new class of antibacterials. In line with this conjecture, it was demonstrated that the *topA66* mutation in *E. coli* led to decreased SOS-response and increased sensitivity to antibiotics[30]. Similarly, overexpression of *M. tuberculosis* TopoI CTD resulted in increased susceptibility to antibiotics and oxidative stress (Banda, Cao and Tse-Dinh, 2017). Thus, we expect that drug-mediated uncoupling of the RNAP:TopoI complex will lead to the fixation of mutations in gyrase genes, decreasing the gyrase activity. If so, administration of such a hypothetical drug followed by a course of gyrase-targeting inhibitor, we speculate, could be therapeutically beneficial. A gyrase inhibitor acting on an essential enzyme whose activity is already decreased by mutations will probably be leaving less space in the mutational landscape for accumulation of additional substitutions conferring resistance to antibiotics.

# Methods

## Strains and plasmids

*E. coli* DY330 *topA*-SPA strain (W3110 *lacU169 gal490 cI857 Δ(cro-bioA) topA*-SPA) with *topA* gene fused with the sequence encoding the SPA tag (purchased from Horizon Discovery Biosciences) was used in EcTopoI ChIP-Seq and Topo-Seq, supercoiling, and toxicity experiments. *E. coli* DY330 (W3110 *lacU169 gal490 cI857 Δ(cro-bioA)*), *E. coli* DH5α, and *E. coli* BW25113 strains were used for amplification of *topA* gene, standard cloning, and *topA* gene editing, respectively. *E. coli* DY330 *rpoC-TAP* (W3110 *lacU169 gal490 cI857 Δ(cro-bioA) rpoC*-TAP) strain with *rpoC* gene fused with the sequence encoding the TAP tag (purchased from Horizon Discovery Biosciences) was used for RNAP-ChIP-Seq experiments. *E. coli* CSH50 *λsfiA::lacZ* reporter strain was used for SOS-response detection[79]. pCA24 was used for cloning and overexpression of EcTopoI 14 kDa CTD, EcTopoI, EcTopoI Y319F mutant, and GFP. pCA24 topA was obtained from *E. coli* ASKA collection[80]. pET28 was used for overproduction and purification of EcTopoI in *E. coli* BL21(DE3). pBAD33 was used for overexpression of *topA-strepII wt* and *topA(G116S/M320V)-strepII* double-mutant. All plasmids were verified by Sanger sequencing.

## Cloning of EcTopoI and EcTopoI 14 kDa CTD and construction of the pCA24 topA Y319F plasmid

The *topA* gene was PCR-amplified from genomic DNA extracted from *E. coli* DY330 (Genomic DNA extraction kit, Thermo Fisher Scientific). To remove the NcoI restriction site at the end of the *topA* gene, two overlapping fragments were generated using the following primers: topA_NcoI_fw + NcoI_mut_rev, NcoI_mut_fw + topA_HindIII_strepII_rev (for primer see Supplementary Table 1). The two fragments were joined by overlap extension PCR using topA_NcoI_fw and topA_HindIII_strepII_rev primers. The resulting PCR product was cloned into pET28 at NcoI and HindIII sites, giving pET28 topA-strep. The C-terminal StrepII tag was introduced with topA_HindIII_strepII_rev primer.

The DNA fragment encoding EcTopoI 14 kDa CTD was amplified from *E. coli* DY330 genomic DNA using TopoA_14_kDa_CTD_BamHI_fw and TopoA_CTD_HindIII_rev primers (Supplementary Table 1) and cloned at BamHI and HindIII sites into pCA24 giving pCA24 14 kDa CTD. To prevent the leakage from the T5-lac promoter (and potential toxicity), all plasmid-transformed strains before induction were grown on media containing 0.5% glucose for catabolite repression.

To construct pCA24 topA Y319F, *topA* fused with strepII-coding sequence was PCR-amplified from pET28 topA_strep plasmid with primers topA_NcoI_fw + topA_strepII_HindIII_rev. The pCA24 backbone was PCR-amplified from pCA24 GFP plasmid with primers pCA24_core_NcoI_rev + pCA24_core_HindIII_fw. The topA_strepII PCR product was cloned into pCA24 between the NcoI and HindIII sites resulting in the pCA24 topA_strepII plasmid. The Y319F mutation was introduced into cloned *topA* by QuickChange method with topA_Y319F_fw + topA_Y319F_rev overlapping mutagenic primers, resulting in the pCA24 topA_strep Y319F plasmid. Briefly, pCA24 topA_strepII plasmid

was amplified with Phusion polymerase, then *E. coli* DH5α cells were electroporated with the linear PCR product with overlapping ends.

## Construction of pBAD33 topA-strepII and pBAD33 topA(G116S/M320V)-strepII plasmids

To generate *topA* double-mutant (G116S/M320V), three overlapping fragments were generated by PCR of pET28 topA-strep plasmid using three primer pairs: topA_XbaI_RBS_fw + topA_G116S_out_rev, topA_G116S_in_fw + topA_M320V_in_rev, and topA_M320V_out_fw +topA_strepII_HindIII_rev. The fragments were fused by overlap extension PCR using primers topA_XbaI_RBS_fw+topA_strepII_HindIII_rev (Supplementary Table 1). The final amplicon treated with DpnI was cloned into pBAD33 at XbaI and HindIII sites, resulting in pBAD33 topA(G116S/M320V)-strepII.

To construct pBAD33 topA-strep, a plasmid backbone of pBAD33 topA(G116S/M320V)-strepII was obtained by digestion with NdeI and HindIII and ligated with a PCR product (topA_NdeI_fw + topA_HindIII_strepII_rev primers, see Supplementary Table 1) digested with the same restriction enzymes and DpnI.

## Genome editing, editing of topA gene

*topA* gene editing in *E. coli* BW25113 chromosome was performed by Lambda-Red recombineering using pKD46 plasmid[81]. Recombination cassettes were obtained by PCR from pKD4 plasmid with primers having flanking regions homologous to the sites of desired recombination: topA_delta_topA66_kanR_F + topA_SPA_kanR_cysB_R, topA_delta_14kDa_kanR_F + topA_SPA_kanR_cysB_R, topA_delta_30kDa_kanR_F + topA_SPA_kanR_cysB_R (Supplementary Table 1). Three versions of *topA* truncations from the 5′-end were obtained: *topA-Δ11kDa*, *topA-Δ14kDa*, and *topA-Δ30kDa*.

## Whole-genome sequencing, identification of mutations

Genomic DNA was isolated from 3 mL of an overnight culture of *E. coli* BW25113 or *E. coli* BW25113 *topA* mutants using GeneJET Genomic DNA purification kit (Thermo Fisher). NGS libraries were prepared using the NEBNext Ultra II DNA Library Prep kit (NEB). DNA sequencing was performed on Illumina MiniSeq with 150 + 150 bp paired-end protocol.

Raw reads were filtered and trimmed with Trimmomatic v0.39[82] and then aligned to the *E. coli* W3110 Mu SGS genome (*E. coli* W3110 genome with the insertion of *cat*-Mu SGS cassette may be downloaded from GEO: GSE95567) using BWA-MEM v0.7.17-r1188[83]. SAM, BAM, and bed files were prepared with samtools v1.10[84] and visualized in IGV v2.7.2[85]. SNPs and short indels were identified with bcftools (v1.10.2) mpileup followed by bcftools call programs. Large-scale chromosomal deletions and region multiplications were inspected by manual analysis of genome coverage depth.

## Toxicity assay of EcTopoI 14 kDa CTD

Toxicity during long-term overexpression of EcTopoI 14 kDa CTD was assessed by counting CFUs of *E. coli* DY330 strain transformed with pCA24 14 kDa CTD, a control plasmid pCA24 GFP, or without a plasmid. Overnight cultures were grown at 37 °C in the presence of chloramphenicol (34 μg/mL) in LB, LB supplemented with 0.5% glucose, or LB supplemented with 1 mM IPTG. Serial dilutions of the cultures were applied on LB plates supplemented with or without corresponding additions (0.5% glucose or 1 mM of IPTG). The CFUs were counted after plates' overnight incubation at 37 °C.

To assess the short-term toxicity of EcTopoI 14 kDa CTD, a culture of *E. coli* DY330 transformed with pCA24 14 kDa CTD was grown in LB supplemented with chloramphenicol (34 μg/mL) and glucose (0.5%) until $OD_{600} = 0.2$. Then the culture was divided, one-half was induced with 1 mM IPTG, and another continued to grow uninduced. After 1 h of growth at 37 °C, serial dilutions were applied on LB plates without the inducer, and CFUs were counted.

To assess the effect of EcTopoI 14 kDa CTD overexpression on culture growth, *E. coli* DY330 pCA24 14 kDa CTD (or transformed with pCA24 GFP or pCA24 topA as controls) was cultivated in 600 mL LB supplemented with chloramphenicol (34 μg/mL) and 0.5% glucose. $OD_{600}$ was monitored every 15 min after the culture inoculation. At $OD_{600} = 0.2$ the culture was bisected, and overexpression was induced in one-half with IPTG (final concentration 1 mM). 50 mL culture aliquots were collected at 0, 30, 60, 90, and 120 min time-points after the induction. Cells were pelleted by centrifugation ($3000 \times g$, 5 min, 4 °C), snap-frozen in liquid nitrogen, and stored at −80 °C for further plasmid extraction.

## Plasmid topology analysis by electrophoresis with chloroquine

Plasmid DNA was extracted with GeneJet Plasmid miniprep kit (Thermo Fisher) from frozen cell pellets (see Toxicity assay of EcTopoI 14 kDa CTD). 300 ng of plasmid was separated by electrophoresis (120 V) in 1% agarose gel in ice-cold TAE buffer supplemented with 5 μg/mL chloroquine and visualized by ethidium bromide staining. Experiments were repeated in three independent biological replicates.

## EcTopoI ChIP-Seq

For EcTopoI ChIP-Seq, 40 mL of *E. coli* DY330 *topA-SPA* strain[86] culture was grown at 37 °C in LB containing 50 μg/mL of kanamycin to mid-exponential phase ($OD_{600}$-0.5–0.7) and was crosslinked by using fresh formaldehyde at the final concentration of 1%, followed by incubation at room temperature for 20 min, with agitation. Crosslinking was stopped by the addition of sterile glycine to a final concentration of 0.125 M. Cells were incubated for 10 min at room temperature and harvested by centrifugation at $4500 \times g$, for 5 min, at 4 °C. Cell pellets were washed 3x with 10 mL of ice-cold PBS, resuspended in 1 mL of FA lysis buffer (50 mM HEPES-NaOH pH 7.5, 1 mM EDTA, 0.1% deoxycholate, 0.1 % SDS, 1% Triton X-100, 150 mM NaCl), and incubated on ice for 10 min. Then, protease inhibitors cocktail (cOmplete ULTRA, Sigma-Aldrich) and RNase A (0.1 mg/ml, Thermo Scientific) were added. Cells were disrupted and DNA was sheared by sonication on ice in a 1.5 mL Eppendorf tube to achieve a range of 100–1000 bp fragments. Lysates were clarified by centrifugation at $10,000 \times g$ for 5 min at 4 °C, and the resulting supernatant was used for further analysis.

For the preparation of Input DNA, the initial lysate (100 μL) was treated with proteinase K (Thermo Fisher Scientific) and decrosslinked by incubation at 55 °C for 4 h. Input DNA was purified using a DNA cleanup kit (Thermo Fisher Scientific), and the DNA fragmentation range was assessed by electrophoresis.

For EcTopoI immunoprecipitation, the initial lysate (900 μL) was diluted with 1 mL of TES buffer (10 mM Tris-HCl pH 7.5, 1 mM EDTA, 250 mM NaCl) and mixed with 80 μL of ANTI-FLAG M2 affinity gel (Sigma-Aldrich). Immunoprecipitation was performed at room temperature on a rotating mixer for 1.5 h. Affinity resin was washed consecutively with the following solutions 1 mL each: TES buffer; twice with TESS buffer (10 mM Tris-HCl pH 7.5, 250 mM NaCl, 1 mM EDTA, 0.1% Tween-20, 0.05% SDS); and TE buffer (10 mM Tris-HCl pH 7.5, 1 mM EDTA), followed by brief centrifugation and removing the supernatant. For proteolysis and de-crosslinking after the washing procedure, affinity resin was diluted with 200 μL of TES buffer, proteinase K (Sigma-Aldrich) was added up to 0.5 mg/mL, and samples were incubated at 55 °C for 4 h. Next, the supernatant was collected by centrifugation ($2000 \times g$, 2 min, 25 °C), and IP-DNA was purified using AMPure XP magnetic beads (Beckman Coulter). Enrichment was assessed with qPCR for several genomic loci (Supplementary Table 1).

For further details, see Supplementary Methods.

ChIP-Seq experiments with inhibition of RNAP with rifampicin (Rif) were performed as described above, except that cells were pretreated with 100 μg/mL Rif for 20 min before fixation with formaldehyde.

For EcTopoI ChIP-Seq experiment with overexpression of EcTopoI 14 kDa CTD, 40 mL of *E. coli* DY330 *topA-SPA* cells harboring pCA24 14 kDa CTD plasmid were grown in LB supplemented with 50 μg/mL kanamycin and 34 μg/mL chloramphenicol until $OD_{600}$-0.2. About 14 kDa CTD expression was induced by adding IPTG (1 mM final concentration), and cultivation continued for 1 h before fixing with formaldehyde. The subsequent steps for sample preparation were the same as described above.

ChIP-Seq experiments with overexpression of 14 kDa CTD followed by treatment with Rif were performed as described above for EcTopoI ChIP-Seq with overexpression of EcTopoI 14 kDa CTD, except that Rif (final concentration 100 μg/mL) was added after 1 h of induction of 14 kDa CTD and cultivation continued for another 20 min.

All EcTopoI ChIP-Seq experiments were performed in triplicates.

DNA sequencing was performed at Skoltech Genomics Core Facility using Illumina NextSeq 75 + 75 bp paired-end protocol with NGS libraries prepared by the TruSeq kit (Illumina).

### *E. coli* RNAP-ChIP-Seq

For RNAP-ChIP, *E. coli* DY330 *rpoC-TAP* strain[86] was grown in 200 mL of LB containing 50 μg/mL of kanamycin to a mid-exponential phase ($OD_{600}$-0.5–0.7) and crosslinked with formaldehyde as described above for EcTopoI ChIP. The subsequent steps for sample preparation were also the same, except for the following modifications. Cell pellets were washed twice with 20 mL of ice-cold TBS buffer (20 mM Tris-HCl pH 7.6, 60 mM NaCl), resuspended in 1 mL of ChIP Lysis Buffer (10 mM Tris-HCl pH 8.0, 50 mM NaCl, 10 mM EDTA, 20% sucrose), containing RNase A (1 μg/mL), supplemented with 100 μg of lysozyme, and incubated at 37 °C for 10 min. The mixture was sonicated (in an ice-water bath) to achieve a maximum yield of 300–500 bp fragments. The lysates were clarified by centrifugation at 13,000 × *g* for 10 min at 4 °C, and the resulting supernatant was used for further analysis.

The Input DNA purification and fragment size analysis were carried out as described above, except that the de-crosslinking step was done at 65 °C for 6 h.

For the preparation of RNAP-ChIP DNA, the lysate (-900 μL) was mixed with 10 μL of IgG-agarose (GE Healthcare) and incubated at 4 °C overnight on a rotating mixer in the presence of Protease Inhibitor Cocktail (Sigma). Affinity resin was washed consecutively with 1 mL each of the Wash buffer-1 (40 mM Tris-HCl pH 7.9, 0.5% Tween-20, 2 M NaCl); Wash buffer-2 (40 mM Tris-HCl pH 7.9, 0.5% Tween-20, 1 M NaCl); Wash buffer 3 (40 mM Tris-HCl pH 7.9, 0.5% Tween-20, 200 mM NaCl); and twice with RIPA buffer (50 mM Tris-HCl pH 7.4, 140 mM NaCl, 1% NP-40, 0.1% deoxycholate, 0.1% SDS). All washing procedures were done for 10 min at 4 °C on a rotary mixer. The RNAP-DNA crosslinks were eluted by incubation with ChIP Elution Buffer (10 mM Tris pH 8.0, 30 mM EDTA, 1% SDS) at 65 °C for 4 h on a shaker at 800–900 rpm. The resulting material was treated with Proteinase K (Thermo Fisher Scientific) (0.2 mg/mL) at 37 °C for 3–5 h, followed by IP-DNA de-crosslinking by incubation at 95 °C for 2 h. The IP-DNA was finally purified using ChIP DNA Cleaning & Concentrator kit (Zymo Research).

A 50–100 ng of IP-DNA or Input DNA were end-repaired by a mix of T4 DNA polymerase (NEB), T4 PNK (NEB), and Klenow DNA polymerase (NEB) and purified by Qiaquick PCR DNA purification kit (Qiagen). The eluted DNA material was A-tailed by Klenow Fragment of DNA polymerase (3′–5′-exo⁻) (NEB) and purified by MinElute PCR purification Kit (Qiagen). Illumina Multiplex Adapters (MPA) were ligated with Quick DNA ligase (NEB), and DNA was purified by AMPure XP beads (Beckman Coulter). The resulting library was separated by agarose electrophoresis with subsequent size selection of DNA bands corresponding to 220 bp, which were excised by Gel X-tracta tool (USA Scientific) and purified with Gel Extraction Kit (Qiagen). The resulting DNA was PCR-amplified (18 cycles) using Phusion polymerase (NEB) and purified by MinElute PCR purification kit (Qiagen).

NGS libraries were prepared using the TruSeq kit (Illumina). DNA sequencing was performed by Illumina HiSeq 50 + 50 bp paired-end protocol at Harvard University, Bauer Core Facility.

### ChIP-Seq data analysis

Reads were prepared and mapped to the reference genome as described above for WGS. For EcTopoI ChIP-Seq data, peak calling was performed with MACS2 v2.2.6[37] with the following parameters: nomodel, Q-value <0.001. For RNAP-ChIP-Seq, motif identification was performed by ChIPMunk v8[56] and visualization by WebLogo[87]. Fold enrichment tracks were further analyzed using custom python scripts (https://github.com/sutormin94/TopoA_ChIP-Seq). Detailed analysis is described in Supplementary Methods.

### *E. coli* total RNA-Seq and data analysis

Total RNA was extracted from 2 mL of *E. coli* DY330 culture exponentially growing in LB to $OD_{600} = 0.6$ using ExtractRNA reagent (Evrogen). Samples were treated with DNase I (Thermo Fisher Scientific) and purified by RNAClean XP beads (Beckman Coulter). Sequencing libraries were prepared without rRNA depletion using NEBNext Ultra II Directional RNA Library kit (NEB) with the following modifications: 10 min of fragmentation and ten PCR cycles. The libraries were sequenced on HiSeq 4000 instrument (Illumina, USA) with a 50 bp-long reads protocol. Initial processing of sequencing data (base-calling) was performed with Illumina software HCS v3.3.76 pre-installed in Illumina HiSeq 4000 with standard parameters. Library preparation and sequencing were performed at Skoltech Genomics Core Facility. RNA-Seq was performed in triplicate.

Reads were prepared and mapped to the reference genome as described above for WGS and ChIP-Seq. RSeQC package was used for FPKM and genes expression level calculation[88].

### Strand-specific EcTopoI Topo-Seq and data analysis

*E. coli* DY330 *topA-SPA* cells transformed with pBAD33 topA(G116S/M320V)-strepII plasmid were grown at 37 °C in LB supplemented with chloramphenicol (34 μg/mL) and 0.5% glucose until $OD_{600} = 0.4$. The 100 mL culture was then divided: one-half was induced by adding arabinose to 10 mM (+Ara), and the other half served as a non-induced control (-Ara). Thirty minutes after the induction, cells were harvested by centrifugation (3000 × *g*), and the cell pellet was frozen in liquid $N_2$ and stored at −80 °C until further processing. The cell pellet was resuspended in 1 mL of Strep-Tactin lysis buffer (50 mM Tris-HCl pH 8.0, 150 mM NaCl) containing protease inhibitors (cOmplete ULTRA, Sigma-Aldrich) and RNase A (0.1 mg/ml, Thermo Scientific). Cells were disrupted by sonication as described for EcTopoI ChIP-Seq. Lysates were clarified by centrifugation at 8000 × *g* for 5 min at 4 °C, and the resulting supernatant was used for further analysis.

Input DNA samples (+Ara-IP and -Ara-IP samples) were prepared as described for EcTopoI ChIP-Seq Input DNA.

For immunoprecipitation of EcTopoI-DNA cleavage complexes (+Ara+IP and -Ara+IP samples), 900 μL of the lysate was mixed with 80 μL of Strep-Tactin Superflow Plus affinity resin (Qiagen) pre-equilibrated with Strep-Tactin lysis buffer containing 0.05% SDS. After 1 h of incubation at 25 °C the resin was washed 3x with Strep-Tactin lysis buffer, and the complexes were eluted with 100 μL of Elution buffer (50 mM Tris-HCl pH 8.0, 150 mM NaCl, 2.5 mM desthiobiotin). A 20 μL-aliquot of the eluate was analyzed by SDS-PAGE. The remaining 80 μL of the eluate was treated with proteinase K (Thermo Fisher Scientific) overnight at 50 °C. The IP-DNA samples were purified using AMPure XP magnetic beads (Beckman Coulter). All Topo-Seq experiments were performed in triplicates.

NGS libraries were prepared using a strand-specific Accel NGS 1S kit (Swift Bioscience) suitable for damaged DNA. DNA sequencing was performed by Illumina NextSeq 150 + 150 bp paired-end protocol.

Library preparation and sequencing were performed at Skoltech Genomics Core Facility.

Reads were prepared and mapped to the reference genome as described above for WGS and ChIP-Seq sequencing data. The number of DNA fragments' 3'-ends was calculated per position (N3E) separately for forward and reverse strands, based on the read alignments stored in SAM files. The tracks were scaled by the total number of aligned reads to normalize the coverage across samples, and the biological replicates were averaged. After that, the -IP tracks (+Ara-IP and -Ara-IP) were subtracted from the +IP tracks (+Ara+IP and -Ara +IP) strand-wise, resulting in +Ara and -Ara tracks, respectively. Finally, the -Ara tracks were subtracted from +Ara tracks strand-wise to obtain the enriched signal. The resultant tracks were further analyzed using custom Python scripts (https://github.com/sutormin94/TopoI_Topo-Seq).

### Strand-specific DRIP-Seq and data analysis

For DRIP-Seq, 50 mL of *E. coli* DY330 *topA*-SPA strain[86] culture was grown at 37 °C in LB containing 50 μg/mL of kanamycin and 0.5% of glucose to mid-exponential phase (OD$_{600}$-0.6). Cells were harvested by centrifugation, and nucleic acids were purified using GeneJET Genomic DNA purification kit (Thermo Fisher) according to the manufacturer's protocol but omitting the RNase A treatment. The extracted nucleic acids were sonicated to obtain -150 bp-long fragments.

About 10 μg of anti-RNA:DNA hybrid antibodies (S9.6, Kerafast, ENH001) were preincubated with 20 μL of Protein-G Sepharose beads (Bialexa) suspended in 70 μL of IP-buffer (10 mM Tris-HCl pH 7.5, 1 mM EDTA, 0.1% sodium deoxycholate, 1% Triton X-100, 140 mM NaCl) for 2 h at 4 °C with gentle shaking. After preincubation, 6 μg of purified sheared nucleic acids (concentrations measured by Qubit 1X dsDNA HS Assay) were added to the beads, and the immunoprecipitation continued for 4 h. Then, the resin was washed consecutively with 1 mL of IP-buffer, 1 mL of IPS-buffer (10 mM Tris-HCl pH 7.5, 1 mM EDTA, 0.1% sodium deoxycholate, 1% Triton X-100, 500 mM NaCl), and twice with 1 mL of Wash-buffer (10 mM Tris-HCl pH 8.0, 1 mM EDTA, 0.5% sodium deoxycholate, 1% Triton X-100, 250 mM NaCl). The washed resin was mixed with 0.5 mg/mL proteinase K (Thermo Fisher) in 200 μL of TES buffer and incubated at 55 °C for 1 h. The nucleic acids were purified from the supernatant using AMPure XP magnetic beads (Beckman Coulter). For control, 6 μg of purified sonicated nucleic acids were treated with 5U of RNase HI (NEB) in RNase HI buffer for 30 min at 37 °C, and the nucleic acids were purified with GeneJET Gel Extraction and DNA cleanup micro kit (General cleanup protocol, Thermo Fisher). The immunoprecipitation of RNase HI-treated sample using S9.6 antibodies and subsequent nucleic acid purification was performed as described above.

DRIP experiments with cells expressing EcTopoI 14 kDa CTD were performed similarly using *E. coli* DY330 *topA*-SPA strain harboring pCA24 14 kDa CTD plasmid. Cultures were grown in LB supplemented with 34 μg/mL chloramphenicol and 0.5% glucose. For overexpression of the 14kDa-TopoI-CTD protein, cells were induced at OD$_{600}$-0.2 with 1 mM IPTG, and the cultivation continued for 1 h.

DRIP experiments with cells treated with Rif were performed similarly, except that 100 μg/mL Rif (Sigma-Aldrich) was added to the cells grown to OD$_{600}$-0.6 for *E. coli* DY330 *topA-SPA* or after 1 h of induction of 14 kDa CTD for *E. coli* DY330 *topA-SPA* transformed with pCA24 14 kDa CTD. Cultivation of Rif-treated cultures continued for another 20 min. All DRIP-Seq experiments were performed in triplicates.

NGS libraries were prepared using a strand-specific Accel NGS 1S kit (Swift Bioscience). DNA sequencing was performed with Illumina NextSeq using 75 + 75 bp paired-end protocol. Preparation of libraries and sequencing were performed at Skoltech Genomics Core Facility.

Reads were prepared and mapped to the reference genome as described above for WGS and ChIP-Seq sequencing data. For samples with overexpression of EcTopoI 14 kDa CTD, reads were also aligned to the pCA24 14 kDa CTD plasmid sequence. Strand-specific coverage for the forward and reverse strands, track scaling, and coverage normalization across all biological replicates were carried out as described above for strand-specific EcTopoI Topo-Seq data analysis. Then, coverage depth for the reverse strand was subtracted from coverage for the forward strand. After the scaling, tracks of control samples (treated with RNase HI) were subtracted from corresponding experimental tracks. Resultant tracks were further analyzed using custom python scripts (https://github.com/sutormin94/E_coli_DRIP-Seq_analysis).

### EMSA

26-mer oligonucleotides (Consensus, Poly-T, and Random) and their complementary probes were synthesized as non-labeled and 5'-Cy5-labeled forms (Syntol) (Supplementary Table 1). For EcTopoI-DNA-binding evaluation, 0.4 pmols of Cy5-labeled probe was mixed with 0–16 pmols of purified EcTopoI (see Supplementary Methods for the purification procedure) in Binding buffer-1 (10 mM Tris-HCl pH 7.5, 50 mM NaCl, 6 mM MgCl$_2$)[58]. The 20 μL-reactions were incubated at 37 °C for 10 min, and the samples were separated by 10% PAGE (acrylamide/bisacrylamide 29:1) in TGB buffer with magnesium (25 mM Tris-HCl, 250 mM glycine, 6 mM MgCl2, pH 8.3) at room temperature at 100 V. Cy5-labeled bands were visualized by ChemiDoc imaging system (Biorad).

For competition-binding experiments, 0.4 pmols of labeled oligonucleotide was mixed with saturating amounts of EcTopoI (16 pmols) and 0–51.2 pmols of non-labeled competitor oligonucleotide (up to 128-fold molar excess over the labeled oligonucleotide) in Binding buffer-1. Reactions and DNA separation were performed as described above.

DNA fragments of *dps*, *potF*, or *nuoN* were PCR-amplified from *E. coli* DY330 genomic DNA (for primers, see Supplementary Table 1) and purified by GeneJET Gel Extraction and DNA cleanup micro kit (PCR cleanup protocol, Thermo Fisher). For a DNA-binding assay, 2.85 pmols of DNA fragment was mixed with 0–16 pmols of purified EcTopoI in 20 μL of Binding buffer-1. Reactions were incubated at 37 °C for 10 min, and the samples were separated in 10% PAGE in TAE buffer at room temperature at 100 V. For DNA visualization, the gel was stained with EtBr. Experiments were performed in triplicates.

### Oligonucleotides cleavage by EcTopoI

For EcTopoI-induced DNA cleavage evaluation in vitro, 0.6 pmols of Cy5-labeled 26-mer probe (Supplementary Table 1) was mixed with 0–12 pmols of purified EcTopoI in Cleavage buffer (Binding buffer-1 without MgCl$_2$) in 15 μL-reactions. As a control, 0.6 pmols of heat-inactivated (10 min, 95 °C) EcTopoI was mixed with a probe in a separate reaction. Reactions were incubated at 37 °C for 45 min and stopped by adding SDS to a final concentration 1%. Samples were treated with 0.5 mg/mL proteinase K (Sigma-Aldrich) at 95 °C for 1 h. DNA fragments were separated in PAGE and visualized as described for EMSA for oligonucleotides. Experiments were performed in triplicates.

### Microscale thermophoresis (MST)

10 fmols of Cy5-labeled 26-mer oligonucleotide (Consensus, Poly-T, Random and complementary sequences) was mixed with purified EcTopoI (0–17 pmols) in 20 μL of Binding buffer-2 (10 mM Tris-HCl pH 7.5, 50 mM NaCl, 6 mM MgCl$_2$, 0.05% Tween-20). Reactions were incubated at room temperature for 5 min, and a 5 μL-aliquot was loaded into NT.115 capillaries (NanoTemper). MST was performed in Monolith NT.115 controlled by MO.Control v2 (NanoTemper) at 22 °C, using excitation power at 100% and MST power at 40%. The binding curve fitting, $K_D$ measurements ($K_D$ model), and statistic evaluations were performed automatically with default parameters using MO.Affinity Analysis v3 software (NanoTemper).

## Microscopy

Induced and non-induced *E. coli* DY330 *topA-SPA*/pCA24 14 kDa CTD cells were grown as described for EcTopoI ChIP-Seq. *E. coli* BW25113 cultures were grown in LB till $OD_{600}$-0.6. Cells were spotted on agarose pads (1.2% agarose in PBS) and imaged at 100x magnification using a Nikon Eclipse Ti microscope was controlled by NIS-Elements BR 4.51.01 and equipped with the Nikon Plan Apo VC 100×1.40 oil objective and Nikon DS-Qi2 digital monochrome camera. Images were processed using ImageJ2 v2.35 software[89]. Experiments were repeated in three independent biological replicates.

## Reporting summary

Further information on research design is available in the Nature Research Reporting Summary linked to this article.

## Data availability

The sequencing data supporting the findings of this study have been deposited in NCBI's Gene Expression Omnibus with corresponding dataset accession numbers for *E. coli* RNA-Seq GSE181687, *E. coli* TopoI ChIP-Seq and Topo-Seq GSE181915 and GSE182473, respectively), *E. coli* RpoC ChIP-Seq GSE182850), *E. coli* DRIP-Seq GSE181945). Sequencing data for *E. coli topA* mutants' whole-genome sequencing was deposited in NCBI Sequence Read Archive (SRA) under the accession number PRJNA757761). See a full list of NGS datasets used in the study in Supplementary Table 2. *E. coli* W3110 genome annotations with ORFs, operons, and TUs were obtained from Ensembl Bacteria[90], DOOR[91], and EcoCyc[92] databases, respectively. Information about the subcellular localization of *E. coli* proteins was retrieved from PSORTdb 4.0[93]. Annotations of promoters and transcription factor sites were obtained from RegulonDB[94]. Source data are provided with this paper.

## Code availability

The custom scripts used for data analysis and visualization have been deposited in GitHub: TopoI_ChIP-Seq https://github.com/sutormin94/TopoI_ChIP-Seq, TopoI_Topo-Seq https://github.com/sutormin94/TopoI_Topo-Seq, E_coli_RNA-Seq_analysis https://github.com/sutormin94/E_coli_RNA-Seq_analysis, E_coli_DRIP-Seq_analysis https://github.com/sutormin94/E_coli_DRIP-Seq_analysis.

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

## Acknowledgements

This work (bioinformatic analysis, microscopy, and microscale thermophoresis) was supported by grant 075-15-2019-1661 from the Ministry of Science and Higher Education of the Russian Federation. This work was also supported by Skoltech NGP Program (Skoltech-MIT joint project) and RFBR grant, project number 20-34-90069, the intramural funds from the Department of Cell Biology and Neuroscience at Rowan University (S. Borukhov), and by the National Institute of Health Grant R01GM130942 (S. Borukhov). Sequencing at Skoltech Genomics Core Facility was supported by the Skoltech Life Sciences Program grant. Genome sequencing of *E. coli topA* derivatives was supported by Russian Science Foundation (Grant 22-14-00004 to O. Musharova). We are grateful to Dr. Marina Serebryakova for mass spectrometry and Dr. Svetlana Dubiley for extensive and fruitful project discussions.

## Author contributions

D.S. conceived the study and designed experiments. A.G. performed EcTopoI ChIP-Seq and DRIP-Seq experiments. D.S. conducted Topo-Seq and RNA-Seq experiments. S.B. and K.O. performed RpoC ChIP-Seq experiments. D.S. constructed *E. coli* BW25113 strains with edited *topA* gene and O.M. prepared sequencing libraries and performed WGS for the strains. D.S. performed all NGS data analysis. A.G. conducted DNA cleavage and DNA-binding experiments. D.S. analyzed the topology of plasmids. A.R. performed microscopy. D.T. conducted pull-down experiments and purified EcTopoI and EcTopoI CTD. D.S. prepared all figures. D.S., S.B., and K.S. wrote the manuscript, which was read, edited, and approved by all authors.

## Competing interests

The authors declare no competing interests.
