## [Peer Review File · Nature Communications]

Interaction Between Transcribing RNA Polymerase and Topoisomerase I Prevents R-loop Formation in E. coliREVIEWER COMMENTS

Reviewer #1 (Remarks to the Author):

Summary

The relationship between E. Coli Topo1 and transcription is the subject of this interesting work by Sutormin et al., which provides important insights about how bacterial RNA polymerase coordinates its function with topoisomerase 1 during transcription. This topic has been previously explored in other systems e.g., mycobacteria and human cells. Despite the structural differences between eukaryotic and prokaryotic enzymes, the findings strongly highlight how the mechanism that ensures coordination between DNA topology and transcription is broadly conserved across multiple domains of life.

The authors show that: i) Topo 1 co-localizes and interacts (via Topo 1 CTD) with the RNA polymerase at transcribed units (TU); ii) Topo 1 also localizes in regions upstream of TSSs where negative supercoils are predicted to accumulate, especially in case of highly expressed and/or divergently oriented TSSs; iii) in the 3' end of the gene, where positive supercoils accumulate, gyrase is found to bind DNA. This is in accordance with the twin domain model. iv) Topo 1 CTD acts as dominant negative as it competes with full length Topo 1 for its interaction with the RNA polymerase. Overexpression of Topo 1 CTD changes the distribution of Topo 1 along the transcribed units.

While I think this is an important advancement in our understanding of transcriptional regulation by DNA topology in prokaryotic organisms, the functional link between Topo 1 and transcription is not sufficiently described. The manuscript could benefit of additional experiments to directly prove the novelty of the authors' claim. The current form of the manuscript also requires a better presentation.

My reservations and specific points of critique are outlined below.

- Figure 3. When overexpressed, Topo 1 CTD competes with Topo 1 full length for RNA polymerase interaction. The main question here is: what is Topo 1 doing along the TU while also interacting with the polymerase? If Topo 1's role is to facilitate transcription by removal of supercoiling then CTD overexpression should induce accumulation of supercoiling, which will in turn impair transcription. To provide functional information about Topo 1's role in TU, the authors should assess the effect of CTD overexpression on RNA polymerase binding by ChIP-seq. Measuring the level of mRNA of a few candidate genes by RT-qPCR (Figure 5SH) is not sufficient to test the effect of CTD overexpression on elongating RNA polymerase (as mRNA levels depend also on the rate of degradation). The fact that there is no increase in Topo 1 binding upstream of TSSs upon CTD overexpression also suggests that transcription is affected, otherwise one would expect more supercoiling upstream of TSS and consequently more Topo 1 binding.

Figure 3F should have the same scale (y axis) as 3B. This will help the reader appreciating the effect of Rif on RNA polymerase and Topo 1. For the same reason, Figures 3G and 3H should be reported in the same scale (y axis) as Figures 3C and 3D, respectively.

- The plots in Figures S6B-D could be complemented by adding a few single gene examples where the increase in Topo 1 binding at IRs of divergent vs convergent genes is clear. This will make the interpretation of the plots easier for the reader.

- The authors assume that the recruitment of Topo 1 in the upstream region of divergent genes is due to accumulation of negative supercoiling, but do not provide direct evidence about this. The authors might want to consider performing psoralen binding assay coupled to qPCR at specific loci to directly detect negative supercoiling (as performed in Herrero-Ruiz Cell Reports, 2021 DOI: 10.1016/j.celrep.2021.108977). Alternatively, Topo-seq can be used as a proxy for

relaxation/detection of negative supercoiling. In this regard, is the signal from the Topo-seq enriched at IRs between divergent highly expressed genes?

- Figure 5C. Topo-seq shows that Topo 1 is not catalytically engaged with the DNA in TU. This finding is puzzling as one might expect that negative supercoiling is generated also in TU (behind an elongating polymerase). In fact, R-loops accumulation in TU upon CTD overexpression is an indication of defects in the relaxation of negative supercoiling (Figure 6H). One potential explanation is that at highly expressed genes, a train of RNA polymerases cancel each other's supercoils, while at medium and low expressed genes the Topo 1 activity is essential for the removal of negative supercoiling. Figure S9G clearly confirm this prediction but it is not cited in the manuscript, nor are the findings described accurately. I suggest generating a simplified version of Figure S9G (for example a single plot comparing Topo-seq at high, medium and low expressed genes) to add to Figure 5 and include a description/interpretation of this result in the text. This might facilitate also the editing of the Discussion (see below), which in my view, could be simplified.

- Discussion could be improved, simplified and shortened. If the authors do not provide direct evidence of negative supercoiling accumulation upstream of TSS they should tone down some statements (for example line 765, line 795 "for the first time", etc). Moreover, chromatin-like remodeling could be responsible for changes in DNA topology upstream of TSS, which trigger Topo 1 recruitment. This hypothesis should be mentioned in the discussion. Whether the RNA polymerase regulates the activity of E. Coli Topo 1 (as the mammalian RNA Pol 2 does upon phosphorylation, which marks the beginning of transcriptional elongation) remains to be proved. The authors suggest that Topo 1 is inactive in the beginning of the TU and activity increases towards the end. However, given that strong accumulation of R-loops is observed in the beginning of the HETU upon CTD overexpression (Figure 6H), it is reasonable to speculate that Topo 1 relaxation is particularly needed towards the 5' end of the TU, somehow contrasting the authors' view. Perhaps steric hindrance is preventing Topo 1 from accessing the DNA in the beginning of TU. This idea could be mentioned.

Other comments

- Kouzine et al. NSMB 2008 and 2013 should be cited when explaining why Topo 1 binds regions upstream promoters (for example line 506).
- The y axis of the metagenome plots indicating "fold change" (or the figure legend) should report that the fold change is relative to the "Input" dataset. Otherwise, "fold change" might be misleading.
- Line 541. S5C refers to His-tagged CTD and S5D refers to SPA-Topo 1. The reference to the figures should be carefully checked. Moreover, I found a bit difficult to follow the pull-down experiments shown in Figure S5C, D, E, F. I suggest adding a scheme describing how the pull-downs were performed
- Line 542. "Overexpression of CTD decreased the amount of RNAP copurified with SPA-tagged EcTopoI from E. coli DY330 (Figure S5F)". The band on the top of the gel is faint, so it is hard to draw any conclusion. How many times has this experiment been performed? Is the reduction reproducible?
- In general, in the text the citation of figures and supplementary figures does not follow a linear order and the reader is forced to go back and forth to find the right figure. The authors might want to re-organize the order of some panels to facilitate smooth reading.
- Line 532 "Consistent with the relocalization of EcTopoI to promoters" I don't think the authors can talk about re-localization. The Rif blocks transcription so the RNA polymerase and Topo 1 are blocked at the TSS and are not able to move into the TU.
- Figure 6H. It is not easy to discriminate 4 curves for the coding and 4 curves for the template strands. The authors might want to test/consider alternative representation of the datasets.

Reviewer #2 (Remarks to the Author):

This manuscript by Sutormin et al. reports very high quality comprehensive results that establish the in vivo significance of TopoI-RNAP interaction in *E. coli*, and provides direct experimental evidence of R-loop accumulation when there is interference of this protein-protein interaction. While the twin-domain model for transcription associated DNA supercoiling proposed by Liu and Wang in 1987 has predicted a role of TopoI in removal of transcription-driven negative supercoiling, the model did not imply direct protein-protein interaction between TopoI and RNAP. The extensive Topo-Seq, DRIP-Seq and ChIP-Seq data reported here also provide for the first-time, a whole-genome validation of the twin-domain model. Furthermore, while the sequence preference of bacterial TopoI DNA cleavage sites have been known from in vitro biochemical studies, this is the first time that in vivo bacterial TopoI cleavage sites have been mapped on the chromosome at the nucleotide level.

This work should be of fundamental interest for scientists who need to consider the dynamics of regulation of DNA supercoiling, transcription activities and genome stability at the whole genome level in the study of cellular processes across all kingdoms of life. There are still weaknesses or inaccuracies in the manuscript with regard to experimental design and interpretation of data that should be corrected. These are listed below:

1. Question concerning the physiological significance of TopoI-RNAP interaction: It has been proposed that the TopoI-RNAP interaction is important for preventing formation of R-loop that may inhibit transcription elongation. Yet it is stated on line 558 that "using RT-qPCR, we showed that CTD overexpression does not affect transcription elongation (Figures S5H-I)." even though CTD overexpression is shown here to decrease the TopoI-RNAP interaction. It is not stated if the decrease in topA RT-qPCR product from CTD overexpression seen in Fig S5H and increase in rpmH-rnpA-yidD-yidC cluster RT-qPCR product following CTD overexpression is statistically significant. The authors should discuss in the main text why topA and rpmH-rnpA-yidD-yidC cluster is used for measurement of transcription elongation. The transcription elongation of rRNA genes has been shown by others to be sensitive to topA mutation. Perhaps synthesis of rRNA should have been examined for effect from the CTD overexpression.
2. Figure S9A is supposed to provide evidence that recombinant protein expression from the pBAD33-EcTopoI G116S/M320V and pCA24-14kDa CTD ones induce the SOS response in *E. coli*. pCA24-GFP clone is included as control. However, it is not clear from the image in Figure S9A that there is significant difference between the transformants of pCA24-14kDa CTD and pCA24 GFP. A control of WT pBAD33-EcTopoI transformant should have been included as control for the pBAD33 mutant EcTopoI clone. Information on CSH50 relevant for it to be an SOS reporter strain has not been included in the manuscript or supporting information.
3. With regard to the use of the EcTopoI G116S/M320V double mutant to map EcTopoI cleavage sites: Line 606 states that "continuous production of EcTopoI G116S/M320V from a plasmid led to growth inhibition (Figure S9B) and then line 610 states that "During the experiment, the expression of EcTopoI G116S/M320V had no apparent adverse effect on cell culture growth (Figure S9B)". Even if the difference in OD between non-induced and induced cultures may be small after 30 min of induction of this mutant TopoI that cannot religate the DNA breaks formed after DNA cleavage, the authors cannot assume that SOS induction from DNA damage caused by the mutant topoisomerase covalent complex and cell death have not already occurred.
4. Line 629: the cleavage activity of EcTopoI was significantly lower at promoters and within the TU bodies than at upstream of TU regions (Figure 5D), i.e., at sites where the formation of complexes with RNAP is expected." The authors suggested on line 634 that "topoisomerase activity may be negatively regulated within the RNAP complexes" but that would decrease the advantage of having the topoisomerase activity associated with the RNAP through PPI. The EcTopoI cleavage sites associated with RNAP complex might be preferentially repaired because of the activity of the Mfd protein if an RNAP is stalled by the trapped EcTopoI cleavage complex.
5. On line 710, the authors proposed that "uncoupling of the RNAP:EcTopoI complex formation by CTD overexpression phenotypically mimics the inactivation of EcTopoI in topA Δ 30 clones". The topA Δ 30 protein is deficient in both relaxation activity and PPI through the CTD. The comparison of CTD overexpression phenotypes with topA gene inactivation should be conducted by examining phenotypes of a strain with EcTopoI inactivated by mutation in the N-terminal catalytic domains.
6. It cannot be ruled out that CTD overexpression perturbs other interactions in vivo besides the

RNA:EcTopoI complex formation. The EcTopoI gene in the chromosome can be replaced with a non-homologous bacterial topoisomerase I to uncouple the RNAP:EcTopoI interaction. Such comparison would better validate the extent of physiological requirement of the direct RNAP:topoI interaction.

7. The authors proposed on line 754 that “uncoupling of RNAP:EcTopoI association leads to accumulation of transcription induced R-loops, which may be the basis of toxicity of overexpressed 14kDa CTD”. This should be tested by testing if the toxicity of overexpressed 14 kDa CTD can be reduced by overexpressing RNase H at the same time.

8. Line 908 states that “We did not observe any compensatory mechanisms in the cell for uncoupling the RNAP:EcTopoI complex. Instead, cells rapidly acquired mutations in the gyrase genes and/or amplified genes of TopoIV”. This is inaccurate because the compensatory mutations and gene amplifications are found in strain with topAΔ30 which lacks both relaxation activity and RNAP interaction. Chromosomal mutations/amplifications that compensate for CTD overexpression have not been described. The argument beginning in line 916 “we expect that drug-mediated uncoupling of the RNAP:TopoI complex will lead to the fixation of mutations in gyrase genes, decreasing the gyrase activity. If so, administration of such hypothetical drug followed by a course of gyrase-targeting inhibitor would be therapeutically beneficial” is problematic. Currently clinically approved gyrase-targeting antibiotics are gyrase poisons that would decrease in efficacy if gyrase activity is decreased because the bactericidal effect of these gyrase-targeting antibiotics requires the accumulation of the gyrase cleavage complex on chromosomes.

Reviewer #3 (Remarks to the Author):

In this paper Dimitry Sutormin and colleagues have mapped TopoI binding sites genome wide in E.coli by using ChIP-Seq. By superpositioning RNAP chip, they show that TopoI is colocalized with RNAP especially in highly transcribed transcription units. As expected, TopoI occupancy is more on divergent gene and promoter regions. Notably they find TopoI recruitment in upstream region of highly expressed genes in addition to colocalization with RNAP. Direct interaction between TopoI and RNAP, earlier shown is further validated with respect to the function of the complex. By combining with their earlier data on genome wide analysis of DNA Gyrase binding, they provide further support for twin supercoiling domain model of transcription, originally proposed by Jim Wang in late '80s. This is a carefully conducted, well executed work addressing the interrelationship between topology and transcription in E.coli.

Although such genome wide studies have been carried out with different species of Mycobacteria, Strptococcus etc, which provided the experimental evidence for the operation of twin supercoiled domains such an in depth study was lacking in E.coli where much of the work on topoisomerases and transcription has been carried out. The presence of back up relaxases and also the absence of inhibitors for TopoI hindered such studies. Surtomin has cleverly overcome this problem by using a poisonous mutant of E.coli TopoI which does not complete the reaction (to assess the genome wide cleavage/ action) and also by over expressing the CTD of TopoI to elucidate the in vivo function of the enzyme. The CTD overexpression leads to R-loops accumulation, indicating that the RNAP:EcTopoI interaction is required for R-loop control. Together, the binding and cleavage analysis of TopoI provide a comprehensive landscape of TopoI action in E.coli, confirming the importance of the enzyme in controlling excess R-loops.

I have a few comments which authors may find useful while revising the manuscript.

1. There are a few over statements in the manuscript which need to be corrected. For example “Taken together, these observations provide, for the first time, a whole-genome validation of the twin-domain model proposed by Liu & Wang”. This is inaccurate representation. This is clearly not the first one as they have cited the previous work. It certainly gives a wrong impression to the readers! The authors may pay attention to such statements elsewhere.

2. Line 514 : These are not two alternate predictions. Both are possible and that is what their results show.

3. Line 527: The increase in RNAP occupancy in LETU upon Rif treatment needs a better explanation , if there is one.

4. Line 596: Gyrase occupancy in convergent genes – Fig 4C . Important to cite previous work (Ahmad et al 2017) .

5. I liked the figure comparing the two variations in twin supercoling domains. Any explanations?

6. Line 915 onwards: I am not sure reduction in Gyase or TopoI activity would help in reduction in mutations and AMR.

Minor comments:

Is Figure one necessary? Perhaps not. If so, a more descriptive legend can be added.

The colour coding in some figures is not very clear. If possible , redrawing some of them is recommended.

RESPONSES TO THE REVIEWERS' COMMENTS

Reviewer #1

This Reviewer stated that our “findings strongly highlight how the mechanism that ensures coordination between DNA topology and transcription is broadly conserved across multiple domains of life.” He/she thought that our work “is an important advancement in our understanding of transcriptional regulation by DNA topology in prokaryotic organisms” but considered that “the functional link between Topo 1 and transcription is not sufficiently described. The manuscript could benefit of additional experiments to directly prove the novelty of the authors’ claim.” There was also an issue about improving the presentation.

Specific points

- Figure 3. When overexpressed, Topo 1 CTD competes with Topo 1 full length for RNA polymerase interaction. The main question here is: what is Topo 1 doing along the TU while also interacting with the polymerase? If Topo 1’s role is to facilitate transcription by removal of supercoiling then CTD overexpression should induce accumulation of supercoiling, which will in turn impair transcription.

To provide functional information about Topo 1’s role in TU, the authors should assess the effect of CTD overexpression on RNA polymerase binding by ChIP-seq. Measuring the level of mRNA of a few candidate genes by RT-qPCR (Figure 5SH) is not sufficient to test the effect of CTD overexpression on elongating RNA polymerase (as mRNA levels depend also on the rate of degradation). The fact that there is no increase in Topo 1 binding upstream of TSSs upon CTD overexpression also suggests that transcription is affected, otherwise one would expect more supercoiling upstream of TSS and consequently more Topo 1 binding.

We appreciate the Reviewer’s idea about the investigating the effect of CTD genome-wide by RNAP ChIP-Seq. However, due to the limited time for revision and budgetary constraints we are unable to perform ChIP-Seq at this time. To address the concern, in the revised manuscript we have included ChIP-qPCR data that show the enrichment of RNAP over two long and active TUs (same as used for RT-qPCR in the original submission) for -CTD and +CTD conditions. The results of this analysis show that indeed, CTD expression decreases RNAP enrichment within the TUs. The new data are shown in **Figures S5G and S5H** in the revised manuscript. The following text was added to describe the findings:

Lines 572-576: “Using ChIP-qPCR for two randomly chosen long (> 2 kb) and medium-to-highly transcribed TUs, we showed that CTD overexpression decreases the enrichment of RNAP toward the TUs ends (**Figures S5G, H**). We propose that this is caused by the premature stalling of transcription elongation complexes due to excessive negative supercoiling.”

- Figure 3F should have the same scale (y axis) as 3B. This will help the reader appreciating the effect of Rif on RNA polymerase and Topo 1. For the same reason, Figures 3G and 3H should be reported in the same scale (y axis) as Figures 3C and 3D, respectively.

We changed Figures 3F and 3B to have the same scaling. However, we do not think that scaling of Figures 3G/3H and 3C/3D is appropriate, as the data represented come from different experiments with different (RNAP versus Topo I) proteins pulled down by different antibodies, meaning that enrichments cannot be compared directly. To compare enrichments in Figures 3C/3G (Topo I) and 3D/3H (RNAP) and

perform statistical analysis, normalization is needed. The normalized data are presented in Figures 3I, 3J, 3K, and 3N and allow readers to appreciate the effects of Rif on both proteins.

- The plots in Figures S6B-D could be complemented by adding a few single gene examples where the increase in Topo 1 binding at IRs of divergent vs convergent genes is clear. This will make the interpretation of the plots easier for the reader.

A new panel (**Figure S6J**) was added showing requested representative regions of the genome. The following legend accompanies this panel:

“(J) Examples showing the increased enrichment of EcTopol in divergent IRs upstream of highly transcribed TUs and decreased enrichment in convergent IRs. Data for four representative genomic regions are shown. Enrichment of RNAP is shown in green; EcTopol - in red; EcTopol in Rif+ condition - in blue; EcTopol in CTD+ condition - in magenta. RNA-Seq data are shown in grey. Data scaling is shown in parenthesis for each genomic track. Locations of IRs with specific enrichment patterns and directions of transcription of flanking genes is shown by black horizontal arrows.”

The following text was added in the main text

Lines 594-595: “Based on these results, we propose that EcTopol is preferentially recruited to regions with excessive negative supercoiling stabilized by local topological barriers (see representative examples in **Figure S6J**).”

- The authors assume that the recruitment of Topo 1 in the upstream region of divergent genes is due to accumulation of negative supercoiling, but do not provide direct evidence about this. The authors might want to consider performing psoralen binding assay coupled to qPCR at specific loci to directly detect negative supercoiling (as performed in Herrero-Ruiz Cell Reports, 2021 DOI: 10.1016/j.celrep.2021.108977). Alternatively, Topo-seq can be used as a proxy for relaxation/detection of negative supercoiling. In this regard, is the signal from the Topo-seq enriched at IRs between divergent highly expressed genes?

We believe that psoralen crosslinking is outside the scope of our own work. However, to follow the Reviewer’s suggestion we can use published genome-wide data obtained with this method. Initially, we tried to use genome-wide psoralen binding data reported in (<https://doi.org/10.1038/ncomms11055>) to compare psoralen binding relative to transcription units and intergenic regions. However, these data seem to be poorly enriched and noisy, so we did not observe any reasonable enrichment that would have indicated accumulation of negative supercoiling upstream of TUs or depletion indicating positive supercoiling downstream of Tus. Analysis of our EcTopol cleavage data generated with Topo-Seq in intergenic regions also was not successful as these data was too noisy and no statistically significant differences in enrichment between different groups of intergenic regions were detected.

Given these unsuccessful attempts, in the present work we use the GapR-Seq data recently published for *E. coli* (DOI: [10.7554/eLife.67236](https://doi.org/10.7554/eLife.67236)). The authors of this paper validated that GapR preferentially binds to positively supercoiled DNA, so enrichment of GapR can be used to assess DNA supercoiling genome-wide. Using these data, we indeed demonstrated that GapR enrichment is decreased upstream of TUs (where negative supercoiling is expected) and increased downstream of TUs (where positive supercoiling is expected) (**Figure 3D**). Moreover, the decrease in GapR enrichment upstream of TUs colocalizes with the increase in EcTopol enrichment and increase in GapR enrichment downstream of TUs colocalizes with the decrease in EcTopol enrichment, as expected. Luckily, while our manuscript was being revised, genome-wide Psora-Seq data were published for *E. coli* (<https://doi.org/10.1093/nar/gkac244>). The authors used biotinylated psoralen to probe negative supercoiling genome-wide. We analyzed their

data with our metagene approach and demonstrated that indeed, signal of Psora-Seq matches the EcTopol enrichment and is increased in upstream regions of active genes, thus addressing the Reviewer's suggestion for which we are thankful.

The following text was added to the manuscript to describe this result:

Lines 607-613: "We used Psora-Seq and GapR-Seq data available for *E. coli* to localize enrichment of topoisomerases with, respectively, regions of negative and positive supercoiling genome-wide. A signal of negative supercoiling revealed by Psora-Seq (Visser *et al.*, 2022) matches the enrichment of EcTopol, indicating that EcTopol accumulation upstream of active TUs indeed colocalizes with increased negative supercoiling. Concordantly, a signal of positive supercoiling revealed by GapR-Seq (Guo *et al.*, 2021) matches the enrichment of DNA gyrase in regions downstream of active TUs (**Figure 3D**)."

- *Figure 5C. Topo-seq shows that Topo 1 is not catalytically engaged with the DNA in TU. This finding is puzzling as one might expect that negative supercoiling is generated also in TU (behind an elongating polymerase). In fact, R-loops accumulation in TU upon CTD overexpression is an indication of defects in the relaxation of negative supercoiling (Figure 6H). One potential explanation is that at highly expressed genes, a train of RNA polymerases cancel each other's supercoils, while at medium and low expressed genes the Topo 1 activity is essential for the removal of negative supercoiling. Figure S9G clearly confirms this prediction but it is not cited in the manuscript, nor are the findings described accurately. I suggest generating a simplified version of Figure S9G (for example a single plot comparing Topo-seq at high, medium and low expressed genes) to add to Figure 5 and include a description/interpretation of this result in the text. This might facilitate also the editing of the Discussion (see below), which in my view, could be simplified.*

This observation was in fact mentioned in the original submission. To emphasize it, we moved the Topo-Seq-related text into a separate section of Results entitled "EcTopol-induced DNA cleavage is increased in upstream regions of active TUs and decreased in TU bodies". A new **Figure 4E** illustrating the decreased cleavage in TU bodies of highly-expressed TUs and increased cleavage in TU bodies of low expressed TUs has been added. The following text has also been added:

Lines 658-663: "Interestingly, while the cleavage is decreased in HETU bodies, it is increased inside LETUs, particularly towards their ends (**Figure 4E** and **Figure S7G**). We speculate that in HETUs, where EcTopol remains inactive, RNAP molecules move in convoys and mutually annihilate positive and negative supercoils (Kim *et al.*, 2019; Alena and Kolomeisky, 2021). In contrast, EcTopol activity is needed to remove excessive negative supercoiling generated by individual RNAP molecules in LETUs."

- *Discussion could be improved, simplified and shortened. If the authors do not provide direct evidence of negative supercoiling accumulation upstream of TSS they should tone down some statements (for example line 765, line 795 "for the first time", etc). Moreover, chromatin-like remodeling could be responsible for changes in DNA topology upstream of TSS, which trigger Topo 1 recruitment. This hypothesis should be mentioned in the discussion. Whether the RNA polymerase regulates the activity of E. Coli Topo 1 (as the mammalian RNA Pol 2 does upon phosphorylation, which marks the beginning of transcriptional elongation) remains to be proved. The authors suggest that Topo 1 is inactive in the beginning of the TU and activity increases towards the end. However, given that strong accumulation of R-loops is observed in the beginning of the HETU upon CTD overexpression (Figure 6H), it is reasonable to speculate that Topo 1 relaxation is particularly needed towards the 5' end of the TU, somehow contrasting the authors' view. Perhaps steric hindrance is preventing Topo 1 from accessing the DNA in the beginning of TU. This idea could be mentioned.*

We tried to improve the Discussion following the Reviewer's suggestions and toned it down. In addition to various edits, the following has been added:

Lines 827-830: "Alternatively, the enrichment of EcTopol in upstream regions might be mediated by transcription-induced chromatin remodeling, though we did not observe any significant skew in enrichment of Fis, HNS, MatP, and MukB nucleoid-associated proteins in these regions (data not shown)."

Lines 832-834: "In contrast, an inverted enrichment pattern is observed for DNA gyrase (by Topo-Seq) and GapR (by ChIP-Seq), proteins known to *i*) act upon/interact with positively supercoiled DNA and *ii*) avoid negatively supercoiled DNA."

Other comments

- Kouzine *et al.* NSMB 2008 and 2013 should be cited when explaining why Topo 1 binds regions upstream promoters (for example line 506).

Suggested papers are now cited, and the following text has been added:

Lines 517-519: "Interestingly, this range is significantly longer than that observed for eukaryotic chromatin, possibly, due to the absence of supercoiling-"buffering" nucleosomes (Kouzine *et al.*, 2008, 2013)."

- The y axis of the metagenome plots indicating "fold change" (or the figure legend) should report that the fold change is relative to the "Input" dataset. Otherwise, "fold change" might be misleading.

The following text was added to all figure legends where enrichments are shown:

"For ChIP-Seq, fold enrichment is given relative to the input sample."

- Line 541. S5C refers to His-tagged CTD and S5D refers to SPA-Topo 1. The reference to the figures should be carefully checked. Moreover, I found a bit difficult to follow the pull-down experiments shown in Figure S5C, D, E, F. I suggest adding a scheme describing how the pull-downs were performed

Figures have been fixed, we are sorry for mistakes in the original submission. To help readers follow the data, cartoon representations of pull-down experiments have been added to Figure S5A (former S5E) and S5C (former S5F).

- Line 542. "Overexpression of CTD decreased the amount of RNAP copurified with SPA-tagged EcTopol from *E. coli* DY330 (Figure S5F)". The band on the top of the gel is faint, so it is hard to draw any conclusion. How many times has this experiment been performed? Is the reduction reproducible?

Indeed, the amount of RNAP co-purified with EcTopol is low (~5% of pulled-down EcTopol). Thus, it is difficult to keep RNAP band visible while avoiding a huge overload of the EcTopol band. We performed more experiments (5 separate pull-downs) and confirmed the reproducibility and significance of the RNAP amount reduction at conditions of CTD overexpression. With new data, the bar-chart on Figure S5D (former S5G) was expanded and standard deviations and p-values included.

- In general, in the text the citation of figures and supplementary figures does not follow a linear order and the reader is forced to go back and forth to find the right figure. The authors might want to re-organize the order of some panels to facilitate smooth reading.

Supplementary figures were thoroughly re-organized and re-numbered to facilitate smooth reading and ensure they are referred to in a linear order.

- Line 532 *“Consistent with the relocalization of EcTopol to promoters” I don’t think the authors can talk about re-localization. The Rif blocks transcription so the RNA polymerase and Topo 1 are blocked at the TSS and are not able to move into the TU.*

We would like to stay with the “relocalization” term, as at least for RNAP the addition of Rif releases substantial amount of RNAP from transcription elongation, effectively increasing the concentration of the enzyme and allowing the binding to weaker promoters which would not have been bound in its absence. We envision that EcTopol “follows” relocalized RNAP in this case.

- *Figure 6H. It is not easy to discriminate 4 curves for the coding and 4 curves for the template strands. The authors might want to test/consider alternative representation of the datasets.*

We tested several alternatives, but they all lead to either increase in the number of curves or number of pictures, so we decided to keep the original picture.

Reviewer #2

This Reviewer stated that our manuscript “reports very high quality comprehensive results that establish the in vivo significance of Topol-RNAP interaction in E. coli, and provides direct experimental evidence of R-loop accumulation when there is interference of this protein-protein interaction.” He/she also acknowledges that “this is the first time that in vivo bacterial Topol cleavage sites have been mapped on the chromosome at the nucleotide level.” He/she noted several “weaknesses or inaccuracies in the manuscript with regard to experimental design and interpretation of data that should be corrected.”

1. Question concerning the physiological significance of Topol-RNAP interaction: It has been proposed that the Topol-RNAP interaction is important for preventing formation of R-loop that may inhibit transcription elongation. Yet it is stated on line 558 that “using RT-qPCR, we showed that CTD overexpression does not affect transcription elongation (Figures S5H-I).” even though CTD overexpression is shown here to decrease the Topol-RNAP interaction. It is not stated if the decrease in topA RT-qPCR product from CTD overexpression seen in Fig S5H and increase in rpmH-rnpA-yidD-yidC cluster RT-qPCR product following CTD overexpression is statistically significant. The authors should discuss in the main text why topA and rpmH-rnpA-yidD-yidC cluster is used for measurement of transcription elongation. The transcription elongation of rRNA genes has been shown by others to be sensitive to topA mutation. Perhaps synthesis of rRNA should have been examined for effect from the CTD overexpression.

Several factors are likely to make transcription sensitive to the topological stress. First, the longer the transcription unit, the more sensitive to topological stress it should be. Second, there must be some level of transcription, which should not be, however, exceedingly high. For example, the rRNA operons, which are among the longest TUs in bacteria, are not good models to assess the sensitivity to topological stress since their extremely high level of transcription implies RNAP molecules convoys which should mutually annihilate the supercoiling effects and also because their stable and highly abundant transcripts are unlikely change significantly upon transient expression of CTD. Based on these considerations, we have chosen to investigate changes in enrichment of RNAP (instead of transcript coverage by RT-qPCR performed initially) over the *topA* and *rpmH-rnpA-yidD-yidC* TUs (each over 2 kbp long and

showing intermediate level of transcription as judged by transcript abundance revealed by RNA-seq). We demonstrated that CTD overexpression decreases the enrichment of RNAP toward the end of these TUs. Probably, this corresponds to premature stalling of RNAP complexes during elongation due to R-loops accumulation and excessive negative supercoiling. **Figures S5G, H** were created to illustrate this phenomenon and the following text was added to the manuscript:

Lines 572-576: “Using ChIP-qPCR for two randomly chosen long (> 2 kb) and medium-to-highly transcribed TUs, we showed that CTD overexpression decreases the enrichment of RNAP toward the TU ends (**Figures S5G, H**). We propose that this is caused by the premature stalling of transcription elongation complexes due to excessive negative supercoiling.”

2. *Figure S9A is supposed to provide evidence that recombinant protein expression from the pBAD33-EcTopol G116S/M320V and pCA24-14kDa CTD ones induce the SOS response in E. coli. pCA24-GFP clone is included as control. However, it is not clear from the image in Figure S9A that there is significant difference between the transformants of pCA24-14kDa CTD and pCA24 GFP. A control of WT pBAD33-EcTopol transformant should have been included as control for the pBAD33 mutant EcTopol clone. Information on CSH50 relevant for it to be an SOS reporter strain has not been included in the manuscript or supporting information.*

Experiments were re-designed and repeated using the suggested additional control. For EcTopol G116S/M320V, **Figure S7B** (former S9A) was updated, and the SOS-response level was quantified. pBAD33 (-) and pBAD33 EcTopol plasmids were added as controls. To demonstrate the 14kDa CTD induces SOS-response, **Figure S8A** was created.

3. *With regard to the use of the EcTopol G116S/M320V double mutant to map EcTopol cleavage sites: Line 606 states that “continuous production of EcTopol G116S/M320V from a plasmid led to growth inhibition (Figure S9B) and then line 610 states that “During the experiment, the expression of EcTopol G116S/M320V had no apparent adverse effect on cell culture growth (Figure S9B)”. Even if the difference in OD between non-induced and induced cultures may be small after 30 min of induction of this mutant Topol that cannot religate the DNA breaks formed after DNA cleavage, the authors cannot assume that SOS induction from DNA damage caused by the mutant topoisomerase covalent complex and cell death have not already occurred.*

Indeed, the overexpression of the intrinsically-poisoned mutant can be deleterious for cells. That is why we induced the EcTopol mutant for a short period of time (30 min). Plating on McKonkey medium (shown in Figure S7B) shows that cells expressing the double EcTopol mutant are undergoing SOS-response (the cell mass in the patches of growing cells is pink colored) yet visually the amount of cells grown from deposited drops of culture aliquots is the same as for the control (no EcTopol mutant expression). The patches of cells shown on the Figure take ~4-6 hours to appear, which suggest that the cells are viable throughout this time. Please recall that the cleavage sites were colocalized with EcTopol ChIP-Seq peaks and negative supercoiling upstream of active TUs, which provides an independent cross-validation and indicates that cell physiology (at least with regards to transcription) was nearly native.

4. *Line 629: the cleavage activity of EcTopol was significantly lower at promoters and within the TU bodies than at upstream of TU regions (Figure 5D), i.e., at sites where the formation of complexes with RNAP is expected.” The authors suggested on line 634 that “topoisomerase activity may be negatively regulated within the RNAP complexes” but that would decrease the advantage of having the topoisomerase activity associated with the RNAP through PPI. The EcTopol cleavage sites associated with RNAP complex*

might be preferentially repaired because of the activity of the Mfd protein if an RNAP is stalled by the trapped EcTopol cleavage complex.

The following text was added to the Discussion to acknowledge the possibility proposed by the Reviewer:

Lines 902-905: “Alternatively, the apparent absence of the TCSs within active TUs may be explained by the activity of transcription-coupled DNA repair pathways which might remove such complexes *in situ* (Park, Marr and Roberts, 2002; Martinez *et al.*, 2022).”

5. On line 710, the authors proposed that “uncoupling of the RNAP:EcTopol complex formation by CTD overexpression phenotypically mimics the inactivation of EcTopol in *topAΔ30* clones”. The *topAΔ30* protein is deficient in both relaxation activity and PPI through the CTD. The comparison of CTD overexpression phenotypes with *topA* gene inactivation should be conducted by examining phenotypes of a strain with EcTopol inactivated by mutation in the N-terminal catalytic domains.

Indeed, the *topAΔ30* mutation might have a pleiotropic effect on the phenotype due to both inactivation of the catalytic activity and the absence of PPI with RNAP. However, it is very difficult to obtain a strain with mutated full-length chromosomal *topA* that encodes a catalytically inactive protein (e.g., with a substitution of the catalytic tyrosine 319 with phenylalanine). *E. coli* strains with catalytically dead Topol rapidly acquire suppressive mutations in gyrase genes and/or amplify genomic regions with TopoIV genes – a well-known phenomenon which we observed for the majority of *topAΔ30* clones. To overcome these limitations, we decided to clone the EcTopol Y319F mutant into the pCA24 plasmid and overexpress it in conditions identical to 14kDa CTD overexpression. Our expectation was that this catalytically inactive protein will outcompete wild-type EcTopol (due to overexpression) and interact with RNAP. The new data included in the Revision show that EcTopol Y319F overexpression is toxic for cells (even more toxic than CTD overexpression), leads to hyper-negative supercoiling of plasmids, and accumulation of R-loops, thus fully matching the CTD phenotype. We, therefore, conclude that EcTopol activity within RNAP:EcTopol complexes is required for normal function of the cell. We wish to thank the Reviewer for proposing this experiment. To accommodate new data **Figures S8F** (Dot-blot), **S8B** (growth curves), and **S8D** (chloroquine phoresis) were added.

The following text was added to revised manuscript:

Lines 714-719: “In order to further characterize the role of topoisomerase activity within the RNAP:EcTopol complex, we overexpressed full-length catalytically inactive mutant EcTopol Y319F from the pCA24 plasmid. Overexpression of this mutant was extremely toxic, presumably indicating the substitution of wild-type, chromosomally-encoded EcTopol with the plasmid-encoded mutant in the complex (**Figure S8B**).”

Lines 743-745: “Overall, we conclude that EcTopol catalytic activity within the RNAP:EcTopol complex is required for optimal growth since its absence phenotypically mimics both the the *topAΔ30* mutation and the deletion of the entire *topA* gene.”

Lines 762-763: “Correspondingly, overexpression of catalytically inactive EcTopol Y319F led to even more dramatic and rapid accumulation of hypercompacted plasmids (**Figure S8D**).”

Lines 786-791: “Dot-blot analysis also demonstrated increased level of R-loops formation in response to CTD or EcTopol Y319F overexpression (**Figures S8E, F**). We conclude that a complex between catalytically active EcTopol and RNAP is required to prevent R-loop accumulation. Transcription-induced R-loops that accumulate upon uncoupling of the RNAP:EcTopol association may be the cause of toxicity observed in the absence of topoisomerase/its activity or when the complex is disrupted by a competitor.”

6. It cannot be ruled out that CTD overexpression perturbs other interactions in vivo besides the RNA:EcTopol complex formation. The EcTopol gene in the chromosome can be replaced with a non-homologous bacterial topoisomerase I to uncouple the RNAP:EcTopol interaction. Such comparison would better validate the extent of physiological requirement of the direct RNAP:topol interaction.

Indeed, we cannot rule out the possibility that CTD disturbs cell physiology through a pathway that is unrelated to EcTopol-RNAP interaction. However, we think that substitution of the *topA* gene with a functional equivalent from another organism could be tricky and is outside the scope of this work. First, to correctly setup an experiment, one should optimize the expression level of a non-homologous topoisomerase to adjust its bulk catalytic activity to that of EcTopol. Second, suppressing mutations in gyrase genes and amplification of TopoIV genes (at least) is expected in such cells should the heterologous topoisomerase activity be imbalanced, which should require continuous monitoring in such a strain. We agree that this experiment is very important for future work. We now mention this caveat in the revised manuscript and would like to address this question in a separate study.

7. The authors proposed on line 754 that “uncoupling of RNAP:EcTopol association leads to accumulation of transcription induced R-loops, which may be the basis of toxicity of overexpressed 14kDa CTD”. This should be tested by testing if the toxicity of overexpressed 14 kDa CTD can be reduced by overexpressing RNase H at the same time.

We tried to express the *rnhA* gene from pBAD30 (Amp resistance, araBAD promoter) alongside with CTD overexpression from pCA24 (Cm resistance, T5-lac promoter), but failed to observe any significant/specific effects. The presence of the pBAD30-*rnhA* plasmid (as well as of the pBAD30-*rnhAD10N* encoding a catalytically inactive RNase HI, or even empty pBAD30) results in a partial decrease in CTD toxicity (1,000-fold compared to a~10,000 fold decrease in viability observed upon overexpression of CTD alone). On the other hand, high level of wild-type *rnhA* expression (induction with 10 mM arabinose) alone was toxic to cells. So stimulatory effects of RNase HI (if any) on transcription-induced R-loop formation are likely hidden by these two factors. We consider this direction as a separate avenue of research for future work.

8. Line 908 states that “We did not observe any compensatory mechanisms in the cell for uncoupling the RNAP:EcTopol complex. Instead, cells rapidly acquired mutations in the gyrase genes and/or amplified genes of TopoIV”. This is inaccurate because the compensatory mutations and gene amplifications are found in strain with *topA*Δ30 which lacks both relaxation activity and RNAP interaction. Chromosomal mutations/amplifications that compensate for CTD overexpression have not been described. The argument beginning in line 916 “we expect that drug-mediated uncoupling of the RNAP:Topol complex will lead to the fixation of mutations in gyrase genes, decreasing the gyrase activity. If so, administration of such hypothetical drug followed by a course of gyrase-targeting inhibitor would be therapeutically beneficial” is problematic. Currently clinically approved gyrase-targeting antibiotics are gyrase poisons that would decrease in efficacy if gyrase activity is decreased because the bactericidal effect of these gyrase-targeting antibiotics requires the accumulation of the gyrase cleavage complex on chromosomes.

We apologize for a misleading statement. By a “compensatory mechanism” we meant a response within the boundaries of “normal physiology” – e.g., changes of expression levels of specific genes, protein modifications, etc., but not fixation of particular mutations. We tested transcription levels of several DNA topology-related genes (*topA*, *topB*, *parC*, *parE*, *rnhA*, *rnhB*, *gyrA*, *gyrB*) by RT-qPCR upon CTD or GFP overexpression and did not observe any specific significant changes.

The text in question was changed:

Lines 953-957: “Except for rapidly acquired mutations in the gyrase and TopoIV genes (DiNardo, Voelkel and Sternglanz, 1982; Pruss, Manes and Drlica, 1982; Brochu *et al.*, 2018), we did not observe changes in expression levels of DNA topology-related genes (*topA*, *topB*, *parC*, *parE*, *rnhA*, *rnhB*, *gyrA*, *gyrB*) upon uncoupling the RNAP:EcTopol complex by CTD overexpression (data not shown).”

Reviewer #3

This Reviewer considered our submission a “carefully conducted, well executed work addressing the interrelationship between topology and transcription in *E. coli*” and offered a few comments.

Specific points

1. *There are a few overstatements in the manuscript which need to be corrected. For example, “Taken together, these observations provide, for the first time, a whole-genome validation of the twin-domain model proposed by Liu & Wang”. This is inaccurate representation. This is clearly not the first one as they have cited the previous work. It certainly gives a wrong impression to the readers! The authors may pay attention to such statements elsewhere.*

The sentence was corrected and now reads as follows:

Lines 834-836: “Taken together, these observations provide a whole-genome validation of the twin-domain model proposed by Liu & Wang.”

2. *Line 514: These are not two alternate predictions. Both are possible and that is what their results show.*

Thank you, indeed, the original sentence was misleading. We changed it to:

Lines 529-531: “In addition, if EcTopol association with extended regions upstream of TUs is driven by excessive transcription-generated negative supercoiling, Rif treatment should abolish this association.”

3. *Line 527: The increase in RNAP occupancy in LETU upon Rif treatment needs a better explanation, if there is one.*

The current estimates of the number of RNAP molecules in the laboratory *E. coli* vary from 11400 ([10.1128/JB.183.8.2527-2534.2001](https://doi.org/10.1128/JB.183.8.2527-2534.2001)) to 4600 ([10.1111/j.1365-2958.2012.08081.x](https://doi.org/10.1111/j.1365-2958.2012.08081.x)) and 3685 ([10.1016/j.bpj.2013.05.048](https://doi.org/10.1016/j.bpj.2013.05.048)) on average per cell. The total number of transcription units in the *E. coli* genome is comparable – 4661 (<https://doi.org/10.1038/nbt.1582>). Therefore, when Rif is added and transcription elongation ceases, all, or nearly all, promoters can be simultaneously bound by an RNAP holoenzyme molecule trapped by Rif, leading to comparable RNAP enrichment across different sites.

The following has been added to make this important point clearer:

Lines 538-542: “This decrease may be caused by the dissipation of transcription-induced negative supercoiling and/or by a more uniform redistribution of RNAP holoenzymes across promoters (since high-affinity promoters cannot be occupied by more than one RNAP molecule, remaining molecules become trapped by Rif at weaker promoters).”

4. Line 596: *Gyrase occupancy in convergent genes – Fig 4C. Important to cite previous work (Ahmad et al 2017).*

This paper is now cited. The revised sentence reads as follows:

Lines 616-619: “Furthermore, while EcTopoI is particularly enriched in IRs flanked by divergent genes (see above) where cumulative negative supercoiling is expected, the DNA gyrase signal is the highest for IRs between convergent genes (where cumulative positive supercoiling is expected) (**Figure 3C**), in line with observations made in *M. tuberculosis* (Ahmed et al., 2017).”

5. *I liked the figure comparing the two variations in twin supercoiling domains. Any explanations?*

Possible explanations for proposed variations in twin domain model can be found in the Discussion section: lines 846-864. The possibilities offered there await their experimental confirmation that would require experiments similar to the ones used in our work in various, possibly non-model, microbes.

6. *Line 915 onwards: I am not sure reduction in Gyrase or TopoI activity would help in reduction in mutations and AMR.*

Indeed, this is a highly speculative assumption, so we toned it down:

Lines 963-967: “If so, administration of such a hypothetical drug followed by a course of gyrase-targeting inhibitor, we speculate, could be therapeutically beneficial. A gyrase inhibitor acting on an essential enzyme whose activity is already decreased by mutations, will probably be leaving less space in the mutational landscape for accumulation of additional substitutions conferring resistance to antibiotics.”

Minor comments

Is Figure one necessary? Perhaps not. If so, a more descriptive legend can be added.

The Figure was moved to supplementary; it is now new **Figure S1**.

The colour coding in some figures is not very clear. If possible, redrawing some of them is recommended.

In the metagene plots the colours are indeed not bright, however, we tried to keep them contrasting and of different intensity, so colorblind persons can see the differences.

REVIEWERS' COMMENTS

Reviewer #1 (Remarks to the Author):

Overall, the revised manuscript has substantially improved, and it is ready for publication in Nature Communications.

I would recommend having the manuscript carefully checked by a native English speaker, as I found it a bit difficult to read in some sections. I also suggest increasing the font of the text in all the supplementary figures, to improve clarity.

Reviewer #2 (Remarks to the Author):

This work validates the physiological significance of EcTopoI-RNAP interaction studied previously in biochemical experiments, and provides a comprehensive picture of the role of EcTopoI and the sites of EcTopo I catalytic activity during in vivo transcription. The additional experiments and revisions have addressed concerns raised in the previous review, and further strengthened the manuscript.

Yuk-Ching Tse-Dinh

Reviewer #3 (Remarks to the Author):

In the revised version, Severinov and coworkers have addressed my comments and suggestions. Now the work is more comprehensive in furthering understanding of topology - transcription interplay in E.coli. I have no further comments.

RESPONSE TO REVIEWERS' COMMENTS

Reviewer #1 (Remarks to the Author):

Overall, the revised manuscript has substantially improved, and it is ready for publication in Nature Communications.

Specific points

I would recommend having the manuscript carefully checked by a native English speaker, as I found it a bit difficult to read in some sections.

The text was checked and revised by a native English speaker.

I also suggest increasing the font of the text in all the supplementary figures, to improve clarity.

Text font was increased in all supplementary figures where possible.

Reviewer #2 (Remarks to the Author):

This work validates the physiological significance of EcTopoI-RNAP interaction studied previously in biochemical experiments, and provides a comprehensive picture of the role of EcTopoI and the sites of EcTopo I catalytic activity during in vivo transcription. The additional experiments and revisions have addressed concerns raised in the previous review, and further strengthened the manuscript.

Yuk-Ching Tse-Dinh

Reviewer #3 (Remarks to the Author):

In the revised version, Severinov and coworkers have addressed my comments and suggestions. Now the work is more comprehensive in furthering understanding of topology - transcription interplay in *E.coli*. I have no further comments.